# Induction of Articular Chondrogenesis by Chitosan/Hyaluronic-Acid-Based Biomimetic Matrices Using Human Adipose-Derived Stem Cells

**DOI:** 10.3390/ijms20184487

**Published:** 2019-09-11

**Authors:** Yijiang Huang, Daniel Seitz, Fabian König, Peter E. Müller, Volkmar Jansson, Roland M. Klar

**Affiliations:** 1Department of Orthopedics, Physical Medicine and Rehabilitation, University Hospital of Munich, 81377 Munich, Germany; yijiang0116@yahoo.com (Y.H.); peter.mueller@med.uni-muenchen.de (P.E.M.); volkmar.jansson@med.uni-muenchen.de (V.J.); 2BioMed Center Innovation gGmbH, 95448 Bayreuth, Germany; dseitz@biomed-center.com; 3Department of Heart Surgery, University Hospital of Munich, 81377 Munich, Germany; Fabian.Koenig@med.uni-muenchen.de

**Keywords:** chitosan, hyaluronic acid, adipose-derived stem cells, biomaterial, articular chondrogenesis, hTGF-β_3_, hBMP-6

## Abstract

Cartilage repair using tissue engineering is the most advanced clinical application in regenerative medicine, yet available solutions remain unsuccessful in reconstructing native cartilage in its proprietary form and function. Previous investigations have suggested that the combination of specific bioactive elements combined with a natural polymer could generate carrier matrices that enhance activities of seeded stem cells and possibly induce the desired matrix formation. The present study sought to clarify this by assessing whether a chitosan-hyaluronic-acid-based biomimetic matrix in conjunction with adipose-derived stem cells could support articular hyaline cartilage formation in relation to a standard chitosan-based construct. By assessing cellular development, matrix formation, and key gene/protein expressions during in vitro cultivation utilizing quantitative gene and immunofluorescent assays, results showed that chitosan with hyaluronic acid provides a suitable environment that supports stem cell differentiation towards cartilage matrix producing chondrocytes. However, on the molecular gene expression level, it has become apparent that, without combinations of morphogens, in the chondrogenic medium, hyaluronic acid with chitosan has a very limited capacity to stimulate and maintain stem cells in an articular chondrogenic state, suggesting that cocktails of various growth factors are one of the key features to regenerate articular cartilage, clinically.

## 1. Introduction

Articular cartilage, a highly organized tissue with substantial durability, features a limited intrinsic self-regeneration potential due to the sparse distribution and limited regenerative capacity of adult chondrocytes, a lack of vascular supply, a slow matrix turnover, and a low supply of progenitor cells [1,2,3,4]. Cartilage lesions occur as a result of trauma, where even minimal damage to the articular cartilage matrix can progress to scar tissue composed of fibrocartilage or age-related degeneration that ultimately leads to osteoarthritis [4,5,6]. Various articular cartilage restoration techniques were developed to try and initiate the repair process. Previous treatments such as joint lavage, periosteum, and perichondrium transplantation including abrasive chondroplasty have become obsolete as they do not regenerate cartilage properly, with recent treatment options focusing more on autologous chondrocyte transplantation and matrix-supported micro-fracturing [4,7,8]. While the newer methods have shown the ability to successfully regenerate cartilage defects, they are temporary in nature, with tissue ultimately degenerating from the required articular cartilage into fibrocartilage with a progression in cumulative damage rather than forming lasting hyaline cartilage and effectively healing the joint [4,7,8]. Thus, regeneration strategies for articular cartilage tissue reformation need to be improved or alternative processes developed that can recapitulate articular cartilage formation, clinically.

Autologous chondrocyte implantation (ACI), a cell-based tissue engineering approach, has been established as a more feasible alternative, providing promising methods for articular cartilage defect treatment [9,10]. The more advanced, matrix-based autologous chondrocyte transplantation requires a combination of cells from a suitable source, a 3-dimensional (3D) biomaterial scaffold and chondrogenic factors [11]. The cell source is one of the most important elements in cartilage tissue engineering. Although chondrocytes can produce the components of the extracellular matrix (ECM), indispensable for fulfilling the functional role of the cartilage, the cells are not available in abundance and have a tendency to lose their phenotype after cellular passaging in vitro. They dedifferentiate into fibroblasts, causing fibrous and mechanically inferior non-articular cartilage tissue formation upon transplantation [12,13,14]. Therefore, alternative cell sources and culture methods have been investigated to overcome these shortcomings, of which mesenchymal stem cells (MSCs) have emerged as the best alternative, as they are present in various tissues such as bone marrow, adipose tissue, cord blood, periosteum, and muscle. These cells can easily be harvested, retain high proliferative capacity, and exhibit multilineage differentiation potential including chondrogenesis [15,16,17]. Bone marrow stromal cells (BMSCs) are commonly used for cartilage engineering but suffer from the same limitations as chondrocytes from autologous cartilage [18,19]. However, unlike BMSCs, adipose-derived stem cells (ADSCs) exhibit obvious advantages including an ease of isolation, relative abundance in autologous adipose tissue, rapidity of expansion, low-donor site morbidity, and multi-differentiation potency [20,21,22,23]. Accordingly, ADSCs should be a promising alternative cell source for articular cartilage tissue engineering. Yet without the appropriate carrier material, cells are limited as to the type of matrix they can form.

Biomimetic, naturally derived porous biomaterials, providing an environment that can spontaneously cause stem cell differentiation or induce a response by the surrounding tissue [24,25,26,27], have taken center stage in tissue engineering research. They support the relevant ECM deposition and stratification, with improved cellular attachment, proliferation, differentiation to the desired phenotype, and integration into the corresponding bone or cartilage defect sites [28,29]. In cartilage-based tissue regeneration, various natural polymers exist, including synthetic alternatives that have been investigated to help in cartilage reformation such as compositions of fiber proteins (collagen, fibrin) [30,31], porous sponge polysaccharides (agarose, alginate, hyaluronic acid, and chitosan) [32,33,34,35], and synthetic hydrogel polymers (polyethylene glycol and polylactic acid) [36,37]. Although synthetic materials allow better control of mechanical, morphological, and physicochemical properties, they more often cause a substantial inflammatory response in vivo than natural materials [38,39]. Chitosan (CHI), a natural biopolymer, is structurally similar to glycosaminoglycans (GAGs), which are a natural component of articular cartilage. Chitosan is highly biocompatible and biodegradable, has no toxic effect, is bacteriostatic, and remains one of the most favored biomaterials for cartilage regeneration [26,27,40]. Previous studies have shown that chitosan or modified chitosan scaffolds promote cartilage matrix formation, reduce inflammation, preserve the phenotypic appearance of chondrocytes, but have a relatively low level of biological activity [41,42,43,44]. However, in some cases, chitosan seems to spontaneously promote non-specific cartilage formation that is not of an articular nature [35,42,45]. According to previous reports, the combination of specific bioactive elements with a natural polymer can generate a device that can enhance the activity of seeded cells and possibly induce the desired matrix formation. Here, hyaluronic acid (HA), a key component in the ECM of articular cartilage and commonly used in biotechnological applications, with similar biological compatibility and biodegradation [34], has also been demonstrated to enhance cellular activities such as migration, proliferation, and differentiation [46,47]. However, the concentration of HA must be confined to a relatively low amount, as high concentrations may reduce cell adhesion because of HA’s negative charge [48]. This disadvantage can be overcome by combining it with a positively charged polycation such as CHI [49], in which a chitosan-hyaluronic-acid-based (CHI/HA) scaffold or injectable gel has been postulated to improve cartilage regeneration in osteochondral defects in vitro and in vivo [50,51,52,53]. Using a 1% hyaluronic acid solution has been shown to reduce inflammatory response and improve cartilage formation [54]. In this context, high-molecular-weight variants have proven more efficient [55]. However, if the regenerated matrix is proper articular cartilage, remains unknown.

Therefore, the present study sought to clarify if a chitosan-hyaluronic-acid-based biomimetic matrix in conjunction with hADSCs does indeed support articular hyaline cartilage formation in relation to a standard chitosan-based construct. This would validate previous in vivo investigations by other research groups [50,51,52,53] and form the base of further investigations with specific morphogens or growth factor combinations to develop a reliable and efficient clinical treatment that can heal articular cartilage damage permanently.

## 2. Results

### 2.1. Characterization of The CHI/HA and CHI Scaffolds

Diffusion and exchange of nutrients and waste throughout the entire scaffold are related to the swelling properties of the scaffolds. Swelling of the CHI/HA and CHI scaffolds are shown in Figure 1A. In the first 4 h, CHI/HA and CHI scaffolds were immersed in DMEM and rapidly increased their weight, reaching a plateau at 8 h. The scaffolds then stabilize, showing only a slight increase with time. Compared with the CHI scaffold, the swelling ratio of the CHI/HA scaffold was significantly higher from the beginning.

It is well known that the degradation of a scaffold is essential in tissue engineering. Thus, the degradability of the CHI/HA and CHI scaffolds was investigated in PBS containing lysozyme in vitro shown in Figure 1B. According to the results, both CHI/HA and CHI scaffolds degraded with culture time, with the former degrading significantly quicker after the fourth week. After 12 weeks, the degradation of the CHI scaffolds was about 50%, with approximately 40% remaining in CHI/HA scaffolds.

### 2.2. Scanning EM of The CHI and CHI/HA Scaffolds Seeded with Differentiating hADSCs

Both the CHI (Figure 2A,B) and CHI/HA (Figure 2C,D) scaffolds appeared as a soft and porous sponge-like disk. The porosity of both scaffolds was very high. SEM micrographs showed that both scaffolds contained pores of approximately 100–200 µm in diameter, with pores being fairly uniformly spaced and possessing a rough morphology when analyzed at higher magnifications (Figure 2B,D). The CHI and CHI/HA scaffolds were seeded with hADSCs, in the fourth passage, and cultured in vitro in modified chondrogenic medium, showing good cell establishment (Figure 2E–H) with differentiated cells seen attaching to scaffolds depositing some fibrous ECM between pores in both scaffolds (Figure 2F,H). At Day 7, cells were observed sitting on a bed of a fibril-like matrix that, through qRT-PCR and immunofluorescent assays, was identified as being cartilage-like. This indicates that most hADSCs had differentiated into chondrocytes and were depositing new ECM, beginning to fill the porous spaces of the devices (Figure 2I–L). By Day 14, cellular numbers increased as witnessed by substantial ECM deposition that was not only observed near the surface of scaffolds (Figure 2M,O) but also seen filling the microporous spaces of the devices (Figure 2N,P). With cartilage matrix formation, again validated by the qRT-PCR and immunofluorescent staining, well advanced by Day 28, the ECM covered all porous spaces (Figure 2Q–T) of the devices.

### 2.3. Proliferation of hADSCs on the CHI and CHI/HA Scaffolds

In order to evaluate whether hADSC numbers were increasing, indicating proliferation, on both the CHI and CHI/HA scaffolds, a WST-1 test and PicoGreen assay were performed 24 h after cell seeding and subsequently after 7, 14, and 28 days of in vitro incubation, respectively. While Pico Green measures DNA content, thus quantifying cell number directly, WST-1 is influenced by cell number and cell vitality. If the results diverge, cell vitality has changed, i.e., vitality per cell has become different. This is not the case here. Both the WST-1 and PicoGreen dsDNA assay (Figure 3A,B) values increased progressively over the 28 day incubation period, indicating a steady increase in cell number in all experimental groups. Starting from the lowest absorbance values and dsDNA quantities measured on the first day of culture, the cell cultures both in the CHI and CHI/HA scaffolds showed a significant increase in their numbers across all-time points of incubation that was significantly higher in chondrogenic (C = chondrogenic medium; CCHI, CCHI/HA) than in normal (N = normal medium; NCHI, NCHI/HA) medium groups (Figure 3A,B). Meanwhile, NCHI to NCHI/HA and CCHI to CCHI/HA groups showed similar patterns in cell number increases with time. The use of chondrogenic medium induced higher cell numbers in CCHI/HA.

### 2.4. Live/Dead Cell Survival Assay on the CHI and CHI/HA Scaffolds

At Day 1, 7, 14, and 28 post seeding, cell survival in CCHI, CCHI/HA, NCHI, and NCHI/HA was assessed using a live/dead staining assay. Human ADSCs were seen attaching to CHI and CHI/HA scaffolds, showing increases in living cell numbers on the surface of the scaffolds over time in both chondrogenic and normal medium groups (Figure 4). However, the cells in chondrogenic medium groups reached higher cell numbers with progressing incubation time than in normal medium. At Day 1 post seeding, viable cells were observed either on the surface or interior of scaffolds including a few dead cells (Figure 4). Cells remained viable throughout the culture period of 28 days, with dead cells only occasionally detected. This indicates that both CHI and CHI/HA scaffolds support the cellular survival of hADSCs.

### 2.5. Immunofluorescence Analyses

Immunofluorescent staining for type II collagen and aggrecan protein expression, markers for general chondrogenesis, was performed to compare matrix formation both in CHI and CHI/HA scaffolds between chondrogenic medium and control groups at Day 7, 14, and 28. Compared to normal medium groups, visual analysis clearly shows significantly more collagen II deposition in the CCHI and CCHI/HA groups. There is also slightly more collagen II present in NCHI/HA than in NCHI scaffolds. While ACAN (Figure 5) was found in lower amounts in normal medium groups on Day 7 and 14, by Day 28 the fluorescent signal appeared to be similar over all treatment groups. Here, qRT-PCR was necessary to reveal variations. The non-collagen matrix protein type I collagen was minimally stained in all four groups (Figure 6), with deposition only increasing slightly by Day 28 of in vitro culture.

### 2.6. qRT-PCR of In Vitro Chondrogenic Differentiation

To evaluate chondrogenic gene expression between the NP, NCHI, NCHI/HA, CP, CCHI, and CCHI/HA groups, relative qRT-PCR gene analysis was performed on in vitro samples, monitoring the relative change in transcription of *ACAN*, *COL1A1*, *COL2A1*, *COL10A1*, *SOX9*, and *COMP*. A pellet control was included in the analysis to see what the effect of morphogens and scaffold types were on the hADSCs. The results represent a snapshot of the above genes at Day 7, 14, and 28 after culturing with modified chondrogenic induction medium or normal medium in CHI and CHI/HA scaffolds or in the form of a 3D pellet. The results have been normalized to four reference genes (*TBP*, *SDHA*, *RPLP0*, *RPL13a*), expressed as log_10_NRQ (normalized relative quantities) including normalization to untreated hADSCs in a monolayer. Relative expression of every gene at different time points but in the same group is shown in Figure 7, whereas the relative expression of every gene in different groups but at the same time point is shown in Figure 8.

*ACAN* expression was found to be upregulated compared to hADSCs growing in the monolayer in all chondrogenic groups including CP, CCHI, and CCHI/HA (Figure 8A) and had increased significantly by Day 28 in these three groups (*p* < 0.001, *p* < 0.0001) (Figure 7A). In cell-scaffold constructs treated with a normal medium, *ACAN* expression was upregulated at Day 7, and then decreased and became downregulated at Day 28 (Figure 7A). Compared to pellet and cell-scaffold groups treated with a normal medium, *ACAN* expression in the chondrogenic medium CP, CCHI, and CCHI/HA was significantly higher at all three time points (Figure 8A). At Day 7 and Day 28, *ACAN* was expressed significantly more in the CCHI/HA group than in the CCHI group. At Day 14, it was higher by trend but not significantly different. The *COL2A1* significantly increased in expression with in vitro culture time in the CCHI and CCHI/HA groups, and decreased significantly in the NP, NCHI, and NCHI/HA groups (Figure 7B). Compared to the NP, NCHI, and NCHI/HA groups, *COL2A1* expression in the CP, CCHI, and CCHI/HA groups were significantly higher (*p* < 0.001) at all three time points (Figure 8B). The expression of *COL2A1* was significantly greater in the CCHI/HA group compared to the CCHI group at Day 7 and 14. At Day 28, COL2A1 was higher in the CCHI/HA group than in the CCHI group, but was not significantly different (Figure 8B).

*COMP* was downregulated in the NP, NCHI, NCHI/HA, and CP groups at all three time points (Figure 7C). The expression of *COMP* in the CCHI group was upregulated at Day 7 compared to hADCS in the monolayer and became downregulated at Day 14 and 28 (Figure 7C). The expression of *COMP* in the CCHI/HA group was upregulated at Day 7 and 14, and then decreased significantly and became downregulated at Day 28 (Figure 7C). Compared to the CCHI and CCHI/HA groups, *COMP* expression in the NCHI and NCHI/HA groups were significantly downregulated (*p* < 0.001) at Day 7, 14, and 28 (Figure 8C). The expression was significantly greater in the CCHI/HA group than in the CCHI group at Day 14. *SOX9* was also upregulated in the CP, CCHI, and CCHI/HA groups at all time points. The expression of *SOX9* in the CCHI and CCHI/HA groups had increased significantly (*p* < 0.01) by Day 14 but decreased at Day 28 (Figure 7D). In the NP, NCHI, and NCHI/HA groups, the *SOX9* expression was upregulated at Day 7 and became downregulated at Day 14 and 28 (Figure 7D). Compared to the NCHI and NCHI/HA groups, the expression of *SOX9* in the CCHI and CCHI/HA groups was significantly higher at all three time points (*p* < 0.001, *p* < 0.0001) (Figure 8D). Between the two chondrogenic medium groups, the expression of SOX9 was significantly greater in CCHI/HA at Day 14 and 28 (*p* < 0.001, *p* < 0.05) (Figure 8D).

*COL1A1* and *COL10A1* were included as negative markers for differentiation towards articular cartilage [55,56], as they are indicators for endochondral bone formation and hypertrophy, respectively. *COL1A1* and *COL10A1* were downregulated in all experimental groups compared to the hADSC monolayer cultures at all three time points (Figure 7E,F and Figure 8E,F), irrespective of scaffold type or culture condition. In pellet cultures of hADSC, downregulation of *COL1A1* and *COL10A1* appears somewhat stronger than in scaffolds, an effect that loses its significance in the light of missing upregulation of positive hyaline markers as well as uniformly low immunohistochemical detection of collagen I in the matrix. While both negative markers remained downregulated, it is interesting to note that the chondrogenic medium did not lead to a higher downregulation of *COL1A1* on Day 7 and 28 (Figure 8E), and even a less strong downregulation for *COL10A1* at all time points (Figure 8F). The gene expression of *ACAN*, *COL2A1*, and *SOX9*, known chondrogenic markers, increased consistently over the 28-day culture period in both cell-scaffold groups with a chondrogenic medium (CCHI & CCHI/HA), but decreased in a normal medium in cell-scaffold constructs (NCHI and NCHI/HA) (Figure 7A,B,D). Gene expression of *COMP*, another chondrogenic marker, was only upregulated in the CCHI group at Day 7 and in the CCHI/HA group at Day 7 and 14 (Figure 7C).

## 3. Discussion

Regeneration of articular cartilage defects remain a challenge in the field of orthopedics due to limited intrinsic healing capacity of this tissue [4]. While it has been demonstrated that the combination of biomimetic scaffolds and autologous stem cells under suitable stimulation can act as an alternative for articular cartilage regeneration [18,28,56], the molecular mechanistic complexities regulating the healing cascade and how to best modulate this to achieve stable hyaline tissue remain problematic [57]. Biomimetic, three-dimensional scaffolds have been shown to reproduce a favorable micro-environment that can provide cell–cell interactions, mechanical and chemical stimulus that together program progenitor cells to more efficiently migrate, proliferate, and differentiate within the superstructure of a biomaterial [38,39]. Chitosan (CHI), a deacetylate derivative from the exoskeleton chitin in insects and crustaceans [58], is a natural polymer with a similar structure to glycosaminoglycan (GAG) that has been widely considered for cartilage repair because it can be modified to create a favorable chondrogenic microenvironment [41,59,60,61,62]. Porous CHI scaffolds have been reported to be a preferable milieu for the proliferation and differentiation of hADSCs compared to micro-mass culture [44]. However, due to its low biodegradation rate and unspecific cell-material interaction, additional supplementation by water-soluble polyanionic biomaterials or natural ECM components are necessary to make it more effective. Indeed, hyaluronic acid (HA), a natural polysaccharide with high biodegradability and biocompatibility, also being part of ECM of articular cartilage and synovial fluid [63], has long been considered as an agent to help supplement chitosan matrices, as hyaluronic acid supports cell proliferation and maintains the chondrogenic phenotype [50,64]. Thus, supplementing the chitosan scaffolds with hyaluronic acid should in theory provide a better environment for progenitor cells to maintain their differentiated chondrogenic phenotype and naturally increase the formation of articular cartilaginous ECM in the scaffold superstructure [46,65].

As suggested previously, in the present study CHI/HA scaffolds showed a better biodegradation than the CHI alone, improving cellular adhesion, survival and ECM deposition [51,65]. The swelling ratio, related to the diffusion and exchange of nutrients, oxygen, and the output of metabolic waste from cells, was greater for CHI/HA scaffolds than for chitosan scaffolds. The degradation rate affects cell growth and cell survival [66], where CHI/HA scaffolds performed better than CHI scaffolds in vitro. Approximately 40% of the CHI/HA matrix is left intact after 12 weeks incubation with lysozyme, revealing that if the material is implanted in vivo*,* it could possibly exhibit a good biodegradation, with sustained mechanical properties and structural integrity until the differentiated cells secrete sufficient ECM that invades the adjacent tissue, establishing a melding of the device with the surrounding cartilage [28]. WST-1 (Figure 2A) and PicoGreen assay (Figure 2B) were performed after cell seeding at 1, 7, 14, and 28 days, in which the cell numbers in the CHI/HA and CHI scaffolds stably increased as culture time progressed, and this increase was significantly greater under the influence of the growth factors from the chondrogenic medium. While it can be argued whether the CHI/HA samples, as compared to the CHI scaffolds, showed a meaningfully better response, as results exhibited similar cell number increases over time between the two culture milieus, the addition of HA in our opinion was supportive of a higher cellular proliferation of hADSCs, an effect previously reported [63,65]. Thus, the live/dead assay results show that hADSCs, albeit thriving in both chitosan scaffolds, grow to greater numbers in CHI/HA scaffolds, independently of the medium used. This suggests that the environment within the scaffolds, particularly in those with hyaluronic acid, favored cellular survival and promoted extracellular matrix formation with time.

The present consensus for stem cell differentiation suggests that differentiation does not depend solely on the exogenous biochemical mediators added to the culture medium, but also the culture conditions employed and the physical interaction between cells, ECM, and/or scaffold [49,67,68,69]. Cell–biomaterial interactions can be established between receptors on the cell surface and adhesion molecules on the surface of materials. According to the literature, it is well known that hADSCs express high levels of the CD44 surface receptor, a glycoprotein with several important physiological functions in cell–cell and cell–matrix interactions including proliferation, adhesion, migration, hematopoesis, and lymphocyte activation [70,71]. Hyaluronic acid is the principle ligand of CD44 and can be formed into HA–CD44 interaction, through which it is assumed to initiate and enhance chondrogenesis of the hADSCs, prevent chondrocytes from dedifferentiating, and promote the subsequent formation of the cartilaginous matrix [34,72,73,74]. Moreover, HA promotes Rho/Rho-associated kinase (ROCK) and phosphatidylinositol-3 kinase (PI3K), two pathways that play essential roles in the regulation of basic cell–cell communication and have a variety of roles in stem cell activity, by binding to the transmembrane receptor CD44 [75,76]. Nevertheless, it has been reported that there are other pathways apart from CD44 mediation involved in HA-initiated chondrogenesis, such as “receptor for HA-mediated motility” (RHAMM) [77], extracellular signal-regulated kinase with SOX9 (ERK/SOX9) [78], and layilin [79]. Hyaluronic acid itself was also used as a cartilage tissue engineering scaffold to control chondrogenesis [80] and to provide a niche for chondrogenic differentiation of stem cells in vitro and in vivo [81]. In the present study, CHI/HA scaffolds with the incorporation of HA supported improved cellular proliferation, survival, and differentiation of hADSCs into the chondrogenic lineage, the deposition of ACAN, and type II collagen protein in the early stage [82,83,84], which possibly was triggered by HA–CD44 interaction. Indeed, as compared to previous studies, our results support the notion that HA can promote hADSCs to differentiate towards a chondrogenic state but does not maintain it, as after Day 7, from our results, the cells seem to de-differentiate if there is no continuous suitable exogenous stimulant.

*Aggrecan*, *COL2A1*, and *SOX9* are markers generally utilized to indicate cartilage formation independent of the type [85,86,87], while *COL1A1*, *COL10A1*, and *COMP* are classical markers to determine the type of cartilage being formed [86,88,89]. Cartilage is distinguished into three types, hyaline cartilage, elastic cartilage, and fibrocartilage according to composition of the ECM. Articular cartilage, in its majority composed of hyaline cartilage, is found on the articular surfaces of joints [90,91]. In the present study, the gene expression patterns of *COL1A1*, a marker for fibrocartilage and *COL10A1*, a marker of hypertrophic chondrocytes in endochondral ossification, were downregulated in all groups (Figure 7E,F), revealing that most matrices being formed in the CHI/HA and CHI treatments were hyaline cartilage to begin with, irrespective of medium type. This was further confirmed by the immunological results where collagen type I immunofluorescence staining (Figure 6) was undetectable in all groups at Day 7, 14, and 28. Articular chondrogenesis is a multi-step process that results in cartilage formation or leads to endochondral ossification during skeletal development [92]. Matsiko et al. (2012) [84] reported that the presence of HA in a collagen-based scaffold improves cellular infiltration and promotes early-stage chondrogenesis for articular cartilage. *SOX9* is one of earliest markers expressed in chondrogenic cell lineage undergoing pre-cartilaginous formation and regulates the expression of *COL2A1* and *ACAN*, which are expressed during mesenchymal cell condensation until differentiation into chondrocytes [93,94]. *SOX9* in this study was only upregulated in CHI/HA and CHI groups under the influence of the chondrogenic medium and only briefly under normal conditions, primarily on Day 7. The concomitant upregulation of *COL2A1* and *ACAN* gene expression, and especially their continued increase up to 28 days in vitro culture are strong indicators of articular cartilage formation, but only take place in the presence of the relevant chondrogenic medium. Under normal conditions, from the present studies results, *COL2A1*, *ACAN*, and *SOX9*, were briefly upregulated at Day 7 in both the chitosan (NCHI) and chitosan/hyaluronic acid (NCHI/HA) groups, suggesting hADSC differentiation was tending towards chondrogenesis. However, by Day 28, *COL2A1*, *ACAN* and *SOX9* were significantly downregulated in the normal medium scaffold groups. In addition, there was little difference in the gene expression patterns between the CHI with and without HA, suggesting two possible causes: 

(1) HA does not support an articular cartilage formation response. Indeed one can question if it does at all improve the chondrogenic reaction at Day 7 as the *ACAN*, *COL2A1* and *SOX9* levels are identical to those of the pure chitosan device.

(2) The decrease in expression after Day 7 of the relevant chondrogenic marker genes would suggest that neither the chitosan nor the hyaluronic acid can maintain an articular chondrogenic state. In the normal medium, *COL1A1* was less downregulated as compared to adherently growing hADSCs in scaffolds, and both *collagen I* and *COL10A1* were less downregulated in the chondrogenic medium. At the same time, *SOX9*, *ACAN*, and *COL2A1* were downregulated in both the chitosan and the hyaluronic groups without a chondrogenic medium. This suggests a tendency to develop fibrous (containing collagen I) and possibly become hypertrophic (with collagen X) cartilage in parallel to increasing chondrogenic differentiation.

The results with respect to the second suggestion as such remain unclear as to what matrix, if any, was being deposited by Day 14 and 28 within the scaffolds in the normal medium. Indeed, only through the addition of our modified chondrogenic medium, with hTGF-β_3_ + hBMP-6 [95], could chondrogenesis be maintained, as *COL2A1*, *ACAN*, and *SOX9* showed constant upregulation for the whole 28 days of culturing, supported by the immunofluorescence data, with typical hypertrophic matrix cartilage specific markers *COL1A1* and *COL10A1* remaining downregulated compared to the hADSCs in the monolayer. Here, ECM deposition commenced at the later stage, *SOX9* being significantly downregulated in both CHI/HA and CHI groups at Day 28, indicating that differentiated hADSCs were maturing and switching to express late stage gene markers of chondrogenesis, particularly *ACAN* and *COL2A1* [94]. It remains unclear as to why downregulation of hypertrophic markers *COL1A1* and *COL10A1* in the normal medium was as strong or even stronger than in the chondrogenic medium, since no concurrent upregulation of positive hyaline markers were observed here. Since both factors remained downregulated as compared to the monolayer culture, to answer the question whether this was the onset of hypertrophic differentiation, protein expression would have to be monitored by immunohistochemistry or Wester-blot over a longer cultivation time than 28 days. For collagen I, an early stage marker of non-hyaline cartilage, no significantly increased expression was found even after 28 days, supporting the hypothesis that the chondrogenic medium improved hyaline differentiation.

While the CHI/HA group on its own under normal culturing conditions does not seem to facilitate proper articular chondrogenesis, when in the presence of the correct signals, as evidence by the modified chondrogenic medium [95], *ACAN*, *COL2A1*, and *SOX9* are upregulated. This suggested that HA synergized with both the hTGF-β_3_ and hBMP-6 to enhance the chondrogenic differentiation of hADSCs into articular chondrocytes that then deposit more articular cartilage matrix within the scaffold, confirmed by the *ACAN* and collagen type II immunofluorescence staining (Figure 6 and Figure 9) in conjunction with the gene expression results (Figure 7 and Figure 8). When compared to the normal medium situation, however, it can be suggested again that the CHI and CHI/HA scaffolds on their own have only limited capacities to direct stem cell differentiation into the articular chondrogenic lineage. The only option left that can recapitulate this would be by following the bone induction principle [96,97], where soluble signals in conjunction with the substratum together bring about proper matrix formation. This was especially made evident in the *COMP* results. The *COMP* gene encodes a pentameric non-collagenous matrix protein expressed primarily in articular cartilage. It has been shown to regulate chondrogenesis and endochondral ossification, and to stabilize the ECM of articular cartilage, by maintaining the structural integrity through its interaction with aggrecan, collagen type II, collagen type IX, and fibronectin [98,99,100,101]. In this study, *COMP* was downregulated as compared to unstimulated hADSCs in the monolayer in all groups except CCHI and CCHI/HA cultured in the chondrogenic medium for 7 and 14 days. During the first 7 days, *COMP* expression was not different between CCHI and CCHI/HA. On Day 14, *COMP* was downregulated significantly in the CCHI group, while in CCHI/HA, it showed a slight increase. This suggests that the HA synergizes with the chondrogenic medium morphogens, thereby enhancing the production of an extracellular cartilage matrix. At Day 28, *COMP* group decreased in its expression pattern in both chondrogenic groups, where we hypothesize that the abundance of matrix molecules possibly leads to a negative feedback loop reducing the expression of specific genes, effectively regulating the chondrocyte metabolism towards an articular matrix [102,103,104].

Comparing growth and matrix formation between normal and growth factor-enhanced medium, the use of morphogens appears to be vital for articular chondrogenesis. The TGF-β isoforms have previously been established to possess good “chondrogenic differentiation potential” when utilizing MSCs [105], as they are present in articular cartilage, and even miniscule quantities of active TGF-β are a potent stimulant for proteoglycan and type II collagen synthesis. Previous studies on the articular cartilage differentiation potential were based on MSCs, or were solely focused on whether TGF-β isoforms also possess similar chondrogenic differentiation capacities in ADSCs [106]. While in vitro TGF-β isoforms stimulated increased *COL2A1*, *SOX9*, *ACAN*, and reduced *COL1A1* expressions [107,108], in-vivo-based research remains problematic. Despite extensive investigations demonstrating the potential of MSCs to regenerate cartilage, the latter degenerates quickly in vivo after a certain number of weeks, leading to ossification rather than maintaining an articular cartilage state [108,109]. Similarly, studies using certain BMP members such as BMP-7, BMP-2, and BMP-4 alone as part of a chondrogenic medium instead of a TGF-β isoform have also generated promising in vitro results [110,111,112,113], yet again, in vivo, the newly formed cartilage structures revert to the typical ossification or fibrocartilage formation [114]. Ude et al. [87] postulated that, by combining two morphogens of the TGF-β supergene family, here hTGF-β_3_ with hBMP-6, an improved formation of persisting hyaline cartilage could be achieved. While Ude et al. [87] utilized stem cells only to generate their theorem, our results to date are the first to suggest that a dual morphogen combination of hTGF-β_3_ with hBMP-6 can rescue articular chondrogenesis even in a scaffold that has issues maintaining it. However, how accurate our results are in relation to previous research, highlighting the positive effects of HA on chondrogenic stem cell differentiation and supposed articular cartilage formation, remains to be elucidated, requiring more investigation to determine whether a CHI/HA scaffold supplemented with hTGF-β_3_ with hBMP-6 and hADSCs is indeed beneficial for long-term articular cartilage regeneration in focal defects or osteoarthritic joints.

## 4. Materials and Methods

### 4.1. Biomaterial Design

Porous sponges were manufactured by lyophilization of glutaraldehyde-cross-linked 0.5% *w*/*v* chitosan hydrogels as described previously [105]. Briefly, chitosan with a 95% degree of deacetylation (Heppe Medical, Halle, Germany) was dissolved at 1% *w*/*v* in 0.1 N HCl with pH 1. Using 1 N NaOH, the pH was carefully adjusted to 5 under constant stirring and dropwise addition of the base. Hydrogels were formed by mixing 1 mL of chitosan solution with 1 mL of an aqueous 1% glutaraldehyde solution (CHI scaffolds) (Sigma-Aldrich, St. Louis, MI, USA). After gelation, samples were frozen at −32 °C using polystyrene insulation to control the freezing rate. Frozen samples were then freeze-dried at −50 °C under vacuum using an Alpha 1–4 LD system (Christ, Osterode am Harz, Germany). For the preparation of a 1% hyaluronic acid for CHI/HA devices, scaffolds were reinfiltrated with a 1% solution of hyaluronic acid sodium salt from Streptococcus equi, MW ~1,500,000–1,800,000 Da (Sigma-Aldrich), and the lyophilization process was repeated. The dry scaffolds were then trimmed on both ends to a final height of 8 mm with a microtomic blade and gamma-sterilized at ca. 27 kGy. Both concentration and molecular weight of hyaluronic acid were chosen according to reported optimal values form the literature [54,55].

The morphologies of CHI and CHI/HA scaffolds were examined using a VHX-5000 3D digital microscope (Keyence, Osaka, Japan) and software VHX-5000 Ver. 1.6.1.0/System Ver. 1.04 (Keyence). The microstructure of the scaffolds was observed by scanning electron microscopy (SEM) (JEOL JSM-6360LV, Tokyo, Japan).

### 4.2. The Properties of CHI/HA and CHI Scaffolds

Swelling of the porous scaffolds was measured according to the following method. The dry weights of the scaffolds (Wd) were measured immediately after vacuum drying for 12 h. Then the scaffolds were immersed in Dulbecco’s modified Eagle’s medium (DMEM) (Gibco, Waltham, MA, USA) maintained at 37 °C and weighed at specific time points (15 min, 30 min, 1 h, 2 h, 4 h, 6 h, 12 h, 24 h, and 48 h) after the removal of excess of water to determine the wet weights (Ww). The swelling ratios of the scaffolds were calculated using the formula: swelling ratio (SR) = (Ww − Wd)/Wd.

Biodegradation of the scaffold were studied according to the literature [44,115]. Briefly, the scaffolds were incubated at 37 °C in phosphate-buffered saline (PBS) (pH 7.4) with 0.5 mg/mL lysozyme (Thermo Fisher Scientific, Waltham, MA, USA) under sterile conditions. The solution was replaced every week by a fresh medium. At the indicted time points (1, 2, 4, 8, and 12 weeks) samples were carefully withdrawn from the medium, thoroughly rinsed with distilled water, and freeze-dried for 24 h to remove excess water. The weight remaining was calculated using the equation: weight remaining (%) = *Wt*/*W0* × 100%, where W0 and Wt are the weights of the scaffolds before and after degradation for 1, 2, 4, 8, and 12 weeks.

### 4.3. Isolation and Culture of hADSCs

Human ADSCs were isolated, as previously described [116], from subcutaneous adipose tissue that was acquired from the Biobank of the University Hospital of Munich Germany which operates in accordance to the European Union compliant ethical and legal framework of the Human Tissue and Cell Research Foundation (http://www.htcr.org). The research was approved by the Human Ethics Committee of the Faculty of Medicine at the University of Munich and the Bavarian State Medical Association. Briefly, harvested adipose tissue was rinsed with phosphate buffered saline (PBS) containing 180 IU/mL penicillin/streptomycin and 0.75 µg/mL amphotericin B (Biochrom, Berlin, Germany), after which the tissue was cut into small fragments and digested with a 0.2 % collagenase A solution (Sigma-Aldrich) in DMEM (Gibco, Waltham, MA, USA) at 37 °C. Afterwards, 15% fetal calf serum (FCS; Sigma-Aldrich) supplemented culture medium was added, after which the mixture was resuspended, filtered through 100 µm sieves, and centrifuged at 400× *g* for 10 min at room temperature (RT). The pellet containing hADSCs was resuspended with fresh growth medium (DMEM, 15% FCS, 60 IU/mL penicillin/streptomycin), seeded in a T-75 culture flask and cultured at 37 °C with 5% CO_2_ for 24 h. Subsequently, the adhered cells were washed with PBS and 20 mL of fresh growth medium was added. The medium was replaced every three days. Human ADSCs used in this study were used at Passage 4.

### 4.4. Cell Seeding Onto CHI and CHI/HA Scaffolds and In Vitro Chondrogenic Differentiation

The dry CHI (control) and CHI/HA scaffolds were placed carefully in a 12-well plate (Thermo fisher scientific, Waltham, MA, USA) and covered with 2 mL of a normal growth medium ((high-glucose DMEM (include 4.5 g/L D-glucose, 110 µg/mL Pyruvate) (Gibco) supplemented with 10% FCS, 60 IU/mL penicillin/streptomycin)). The chitosan and CHI/HA scaffolds were incubated at 37 °C, 5% CO_2_ for 6 h after which the medium was changed and left to incubate overnight at 37 ℃, 5% CO_2_. Thereafter, the medium was removed and the scaffolds were transferred to a new well plates, dried for 3 h before being seeded. Human hADSCs (~90% confluent) were digested with trypsin/EDTA and counted, after which cell suspensions with a concentration of ~1 × 10^7^/mL using a net volume of 100 µL per scaffold were added directly to the biomimetic matrices. Subsequently, cell adhesion was permitted to occur for 2 h at 37 ℃, 5% CO_2_. Two milliliters of a normal growth medium were then carefully added to each well with CHI and CHI/HA scaffolds, which were then allowed to incubate overnight. The following morning (Day 1), a new 12-well plate was prepared, and all the experimental scaffolds with seeded cells were transferred into new wells with 2 mL of either a normal growth medium (Normal + CHI Scaffold = NCHI, *n* = 5; Normal + CHI/HA Scaffold = NCHI-HA, *n* = 5) or a modified chondrogenic medium (Chondrogenic + CHI Scaffold = CCHI, *n* = 5; Chondrogenic + CHI/HA Scaffold = CCHI-HA, *n* = 5). The chondrogenic medium consisted of the normal growth medium supplemented with 10 ng/mL recombinant human TGF-β_3_ (R&D Systems, Minneapolis, MN, USA), 10 ng/mL recombinant human BMP6 (R&D Systems), 100 nM dexamethasone (Sigma-Aldrich), 50 µg/mL L-ascorbic acid-2-phosphate (Sigma-Aldrich), 40 µg/mL L-proline (Sigma-Aldrich), and ITS+1 (Sigma-Aldrich). Final concentrations of 10 mg/L insulin, 5.5 mg/L transferrin, 4.7 µg/mL linoleic acid, 0.5 mg/mL bovine serum albumin, and 5 μg/L selenium were chosen, as previous research has shown that this combination facilitates a better chondrogenic response [85,117,118]. The cell-seeded scaffolds cultured with the normal growth medium was the control group (NCHI and NCHI-HA). The medium was replaced every 3 days, and the progression of chondrogenesis was monitored at Day 7, Day 14, and Day 28. Acellular blank scaffolds were also prepared and incubated in identical conditions to see if the medium altered the porous superstructure of the material.

### 4.5. Pellet Culture and Chondrogenic Differentiation

To have a comparable, standard, three-dimensional (3D) pure cellular scaffold control group, a cell pellet was included in the study [108] to monitor the effect of the medium on the stem cell differentiation. For this, hADSCs were resuspended at a concentration of 2.5 × 10^5^ cells per mL in a normal growth medium ((high-glucose DMEM (include 4.5 g/L D-glucose, 110 µg/mL Pyruvate) (Gibco) supplemented with 10% FCS, 60 IU/mL penicillin/streptomycin)), and 2 mL of the cell suspension containing 5 × 10^5^ hADSCs were transferred into a 15 mL polypropylene conical tube and centrifuged at 500× *g* for 5 min to allow for 3D cell pellet formation. The 3D pelleted cells were then incubated at 37 °C at 5% CO_2_ with loosened caps to permit gas exchange overnight, from which spheroid aggregates formed at the bottom of each tub. The next day (Day 1), the culture medium was replaced with 2 mL of new normal growth (Normal + Pellet; NP; *n* = 5) or the modified chondrogenic medium (chondrogenic + Pellet; CP; *n* = 5) carefully so as not to re-suspend the cell pellet. The 3D pellet medium was changed every 3 days, and 3D cell pellets were cultured for 7, 14, and 28 days prior to harvest and quantitative reverse transcription real-time polymerase chain reaction (qRT-PCR; NP *n* = 5; CP *n* = 5) for analysis. Cells cultured in the normal growth medium were the control group.

### 4.6. Scanning Electron Microscopy (SEM)

In order to see the matrix development progression of hADSCs on CHI and CHI-HA treated with the modified chondrogenic medium (CCHI, CCHI-HA), a device was randomly chosen and cultured for 1, 7, 14, and 28 days. Upon harvest, the cell-scaffold constructs were washed with PBS and fixed in 2.5% glutaraldehyde in PBS overnight at 4 °C. The constructs were then stained with 1% osmium tetroxide, dehydrated in a graded series of alcohols, dehydrated using the critical point drying method, and coated with gold. The samples were examined with a scanning electron microscope (SEM) at an accelerating voltage of 20 kV (Carl Zeiss EVO LS 10, Oberkochen, Germany).

### 4.7. Cell Number/Proliferation Assay

The increase in cell numbers with time indicating proliferation and the metabolic activity of hADSCs cultured in the CHI and CHI/HA scaffolds were evaluated by means of Quant-iT^TM^ PicoGreen dsDNA Kit (Invitrogen, Carlsbad, CA, USA) and a water-soluble tetrazolium-1(WST-1) reagent (Roche, Basel, Swiss), respectively, at Day 1 and subsequently at Day 7, 14, or 28. Briefly, the CHI and CHI/HA scaffolds with the hADSCs were transferred to a new 24-well plate and washed twice with PBS, after which 0.5 mL of a fresh normal growth medium containing WST-1 at 10:1 (*v*/*v*) was added to each well and incubated for 3 h at 37 °C at 5% CO_2_. The absorbance of the WST-1/medium mixture was read at 450 nm using a Synergy HT microplate reader and Gen 5 2.03 software (BioTek, Winooski, VT, USA) in a 96-well plate. The same scaffolds were used for the PicoGreen dsDNA Assay. Here, the 1% PSCs were washed twice with PBS. According to the manufacturer’s protocol, the cells were lysed from the scaffold and DNA standards were mixed with TE-buffer and subsequently with Quant-iT PicoGreen dsDNA reagent. The samples were excited at 480 nm and the fluorescence emission intensity was measured at 520 nm using a Synergy HT microplate reader and Gen 5 2.03 software (BioTek).

### 4.8. Cell Survival in The Scaffold

The effects of CHI and CHI/HA scaffolds on cell survival in both the chondrogenic differentiation medium and the normal growth medium were studied by using a live/dead assay. At Day 1, 7, 14, and 28, the cell-scaffold constructs were stained with a LIVE/DEAD Viability/Cytotoxicity Kit (Invitrogen). Briefly, cell-scaffold constructs were rinsed with PBS and incubated in a staining solution containing Calcein AM and Ethidium homodimer-1 (EthD-1) at room temperature for 30 min, followed by washing with PBS. The constructs were then examined by fluorescence microscopy (Carl Zeiss). Healthy cells fluoresced green, while the nucleus of dead cells fluoresced red [111].

### 4.9. Immunofluorescence Staining

After 7, 14, and 28 days of culture, the cell-scaffolds and 28-day 3D cell pellets were fixed in 4% paraformaldehyde for 30 min at room temperature. The CHI and CHI/HA scaffolds with hADSCs were dehydrated through a graded series of alcohols into paraffin, whereas the 3D cell pellet cultures were embedded in a Tissue-Tek O.C.T.™ compound (Sakura Finetek Germany, Staufen, Germany) and frozen in liquid nitrogen. Following this, 10 µm thick sections were cut either using a Microtome (Leica, Wetzlar, Germany), for paraffin specimens, or CM 3050 cryomicrotome (Leica), for the cryogenic embedded specimens.

To determine the matrix composition secreted by cells within the chondrogenic medium and normal medium groups, immunofluorescence staining for collagen I, collagen II, and aggrecan was investigated. Briefly, paraffin sections from specimens were incubated with primary antibodies (all from Abcam, Cambridge, UK) for either collagen type I (1:300; Cat# ab34710), collagen type II (1:200; Cat# ab34712), or aggrecan (1:300; Cat# ab3778) at 4 °C overnight. The antibodies were diluted with an antibody dilution buffer (Abcam). For negative controls, the first antibody was omitted. The slides were then incubated with the conjugated secondary antibody (Abcam) for 1 h at room temperature. Nuclei of cells were then stained for 8 min with Hoechst 33342 (Life Technologies, Carlsbad, USA). The slides were mounted with Fluoromount W (Serva Electrophoresis, Heidelberg, Germany), air-dried, and stored in darkness at 4 °C. Fluorescence microscopy was then performed with a Zeiss Axioskop 40 equipped with appropriate filter sets and AxioCam MRc 5 (Carl Zeiss). Images were obtained with Axio Vision, Rel. 4.9 (Carl Zeiss). Exposure time was kept constant for the samples where fluorescence intensity was to be compared.

### 4.10. Quantitative Real-time PCR (qRT-PCR)

Quantitative RT-PCR, according to the MIQE guidelines [119], was performed to determine the relative expression of the chondrogenic genes, *aggrecan* (*ACAN*), *collagen type II* (*COL2A1*), *cartilage oligomeric matrix protein* (*COMP*), and *SRY-box 9* (*SOX9*), with *collagen type I* (*COL1A1*) and *collagen type X* (*COL10A1*) being incorporated to determine if cartilage matrix development was purely articular or progressing towards an endochondral ossification lineage. After 7, 14, and 28 days, total RNA was isolated using a modified RNA Trizol extraction procedure [120]. Briefly, 1 mL of Trizol (Invitrogen) was added to the cell material, after which chloroform (Sigma-Aldrich) was added to permit separation of the RNA from the proteinaceous material. After centrifugation, the aqueous RNA containing phase was transferred to a fresh tube where the RNA was then precipitated out by adding isopropanol (Sigma-Aldrich). After incubation at RT for 10 min samples were centrifuged at 16000 rpm over night at 4 °C, upon which RNA pellets were then washed with 75% ethanol (Merck, Billerica, MA, USA) and permitted to dry briefly to prevent alcohol contamination. After drying, total RNA was resuspended in 32 μL of RNase free water (Gibco), after which the concentration and purity of the RNA was determined using a NanoDropTMLite spectrophotometer (Thermo Scientific), and quality was assessed with a Bioanalyzer 2100 (Agilent Technologies, Santa Clara, CA, USA). After RNA extraction, approximately 1 µg of RNA was reverse-transcribed into complementary DNA (cDNA) utilizing the QuantiTect Reverse Transcription cDNA Synthesis Kit (Qiagen, city, Germany).

Quantitative RT-PCR was then performed in duplicate, using the FastStart Essential DNA Green Master (Roche,) on a Light Cycler 96 thermocycler (Roche). Each reaction mixture contained 10 ng cDNA, 10 µM of each primer (Table 1), 2× FastStart Essential DNA Green Master, and RNase-free water to a final reaction volume of 20 µL. The primers of six target genes were designed using Gene Fisher v. 2.0 (http://bibiserv.techfak.uni-bielefeld.de/genefisher2) and optimized according to the MIQE Guidelines [119,121]. Use of GeNorm (http://medgen.ugent.be/~jvdesomp/genorm/) established that *TATA sequence binding protein* (*TBP*), *succinate dehydrogenase complex flavoprotein subunit A* (*SDHA*), *ribosomal protein lateral stalk subunit P0* (RPLP0), and *Ribosomal Protein L13a* (*RPL13a*) were the most appropriate internal reference genes to use in this experiment. All amplified PCR products underwent sequencing (GATC Biotech, Constance, Germany) to confirm that the correct sequence had been amplified. Quantitative RT-PCR cycling conditions included a 3 min pre-incubation period at 95 °C, followed by a three-step amplification program of 40 cycles consisting of a denaturation, annealing, and extension step set at 95 °C for 10 s, 58 °C for 15 s, and 72 °C for 30 s, respectively. The six target genes’ relative expression between samples was normalized to the four reference genes using the qbase+ software (https://www.qbaseplus.com). Data was further normalized to untreated hADSCs in the monolayer.

### 4.11. Statistics

Data are presented as means ± standard deviation (SD, *n* = 5) for the results of WST-1, PicoGreen, and qPCR. qbase+ software was used to analyze the data from qPCR. Microsoft Excel and Prism 5.02 software (GraphPad Software, San Diego, USA) were used for analyzing the data. An ANOVA and a Kruskall–Wallis test were performed to test for the overall effect of variance using R software version 3.6.1 (SAS, Marlow, UK). For positive results, the Student’s *t*-test was used post hoc for comparing groups of data. *p* < 0.05, *p* < 0.01, and *p* < 0.001 were considered significant, highly, and extremely significant, respectively. Statistical significance is indicated by * for *p* < 0.05, ** for *p* < 0.01, and *** for *p* < 0.001.

## 5. Conclusions

Hyaluronic acid has long been appraised for its ability to direct articular chondrogenesis. By combining this substance with chitosan, which has also been shown to possess chondrogenic inducing capabilities, the study sought to clarify if the addition of HA would support the correct matrix formation by differentiating hADSCs to become articular in nature. The chitosan with hyaluronic acid, compared with the chitosan material alone, has an improved swelling capacity, thereby supporting stem cell migration, survival, and differentiation. This suggests that such a device, if utilized in vivo, could be better integrated into an articular defect site than chitosan. However, to facilitate this, the correct molecular signals must be present, as hyaluronic acid with chitosan is insufficient at properly inducing articular chondrocyte differentiation from stem cells. To maintain and improve the regenerative articular cartilage induction, synergistic growth factors need to be added, thereby preventing reversion to fibrocartilage or endochondral ossification. Under these conditions, the chitosan/hyaluronic acid matrix could possibly become a viable alternative to promote better articular cartilage regeneration clinically.

## Figures and Tables

**Figure 1 ijms-20-04487-f001:**
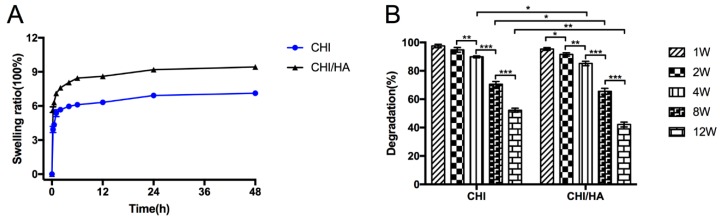
Swelling ratio in DMEM medium at 37 °C (**A**) and degradation in 37 °C warm PBS solution with 0.5 mg/mL lysozyme (**B**) of chitosan and chitosan with hyaluronic acid scaffolds at different time points. (* *p* < 0.05, ** *p* < 0.01, *** *p* < 0.001).

**Figure 2 ijms-20-04487-f002:**
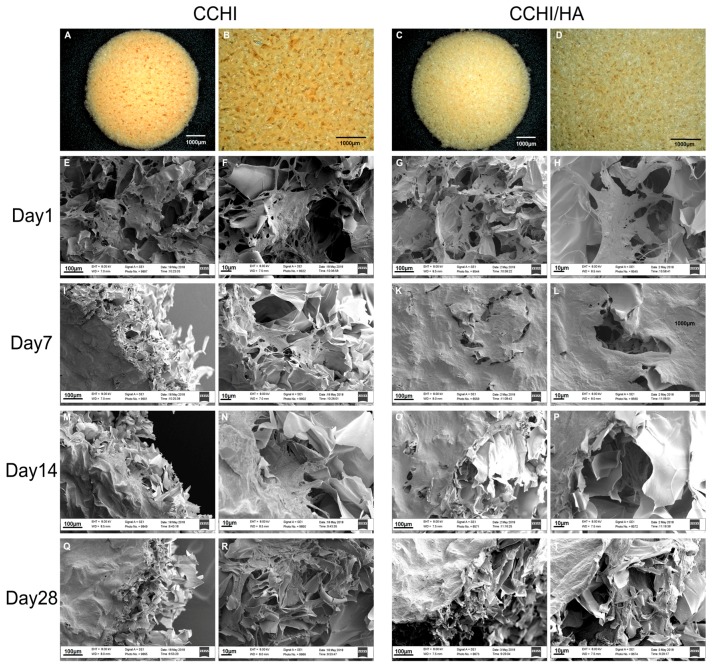
Scanning electron microscopy images (**E**–**T**) of chitosan scaffolds with or without hyaluronic acid at different time points in chondrogenic medium culture, with less magnified overview (**E**,**I**,**M**,**Q** and **G**,**K**,**O**,**S**, respectively) and detail images in higher magnification (**F**,**J**,**N**,**R** and **H**,**L**,**P**,**T**, resp.). The sponge-like topography of non-cultured chitosan scaffold (**A**,**B**) and chitosan with hyaluronic acid scaffold (**C**,**D**) discs is shown before submersion into the medium. After 24 h in the chondrogenic medium with hTGF-β_3_ + hBMP-6, hADSCs were already well established and started to form a matrix (**E**–**H**). Human ADSCs in both scaffold types treated with the chondrogenic medium were observed to quickly and efficiently deposit substantial amounts of a fibrous matrix at Day 7 (**I**–**L**), filling up the microporous structures of the scaffolds. The matrix was aggregating into a woven fibrous structure by Day 14 (**M**–**P**). By Day 28, microstructures of the scaffold material could not be detected by SEM since the scaffolds were completely covered by ECM-like material (**Q**–**T**). Magnifications were set at 300× (**E**,**G**,**I**,**K**,**M**,**O**,**Q**,**S**), 1100× (**F**,**H**,**J**,**L**,**N**,**P**,**R**,**T**).

**Figure 3 ijms-20-04487-f003:**
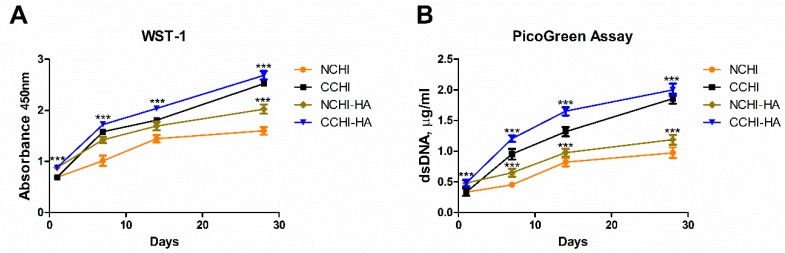
(**A**) WST-1 and (**B**) PicoGreen assays to measure cell number increases (proliferation) for hADSCs on chitosan and chitosan with hyaluronic acid scaffolds cultured in normal (NCHI, NCHI/HA) or chondrogenic (CCHI, CCHI/HA) media. (* *p* < 0.05, ** *p* < 0.01, *** *p* < 0.001)

**Figure 4 ijms-20-04487-f004:**
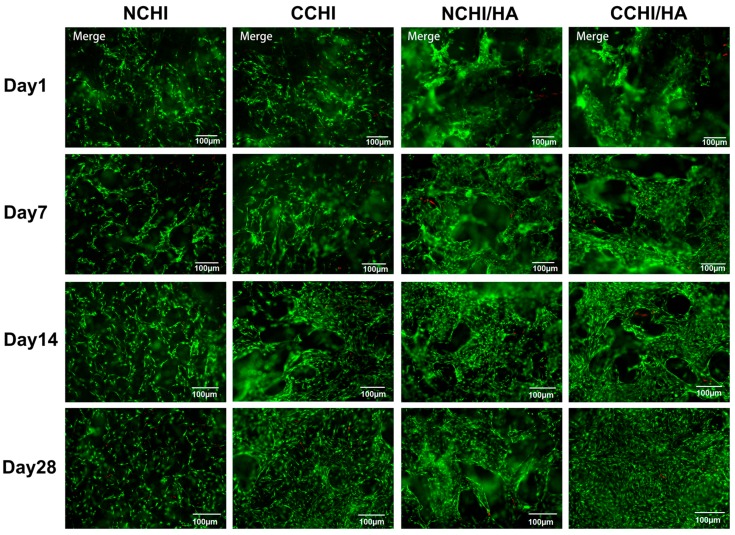
Fluorescent microscopic images of the live/dead cell survival assay in CHI and CHI/HA groups cultured for 1, 7, 14, and 28 days in either N = normal or C = chondrogenic medium. Living cells fluoresce **green** and dead cells fluoresced **red**. Magnification set at 10×.

**Figure 5 ijms-20-04487-f005:**
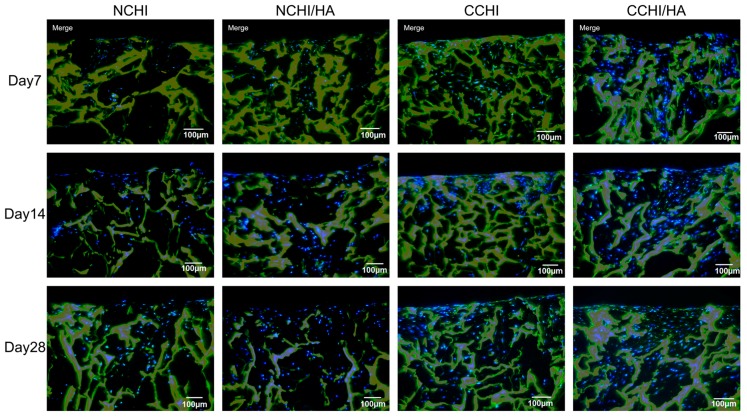
Immunofluorescence staining of aggrecan (**green**) at Day 7, 14, and 28 both in chitosan and chitosan with hyaluronic acid scaffolds with hADSCs cultured in normal (NCHI and NCHI/HA) or chondrogenic medium (CCHI, CCHI/HA). The chitosan and chitosan with hyaluronic acid scaffolds fluoresced **yellow** with living cell nuclei fluorescing **blue**. Magnification set at 10×.

**Figure 6 ijms-20-04487-f006:**
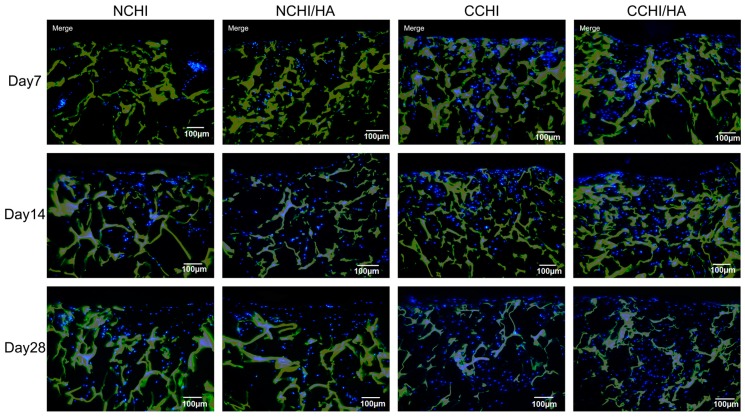
Immunofluorescence staining of collagen type I (**green**) at Day 7, 14, and 28 both in chitosan and chitosan with hyaluronic acid scaffolds with hADSCs cultured in normal (NCHI and NCHI/HA) or chondrogenic medium (CCHI and CCHI/HA). The chitosan and chitosan with hyaluronic acid scaffolds fluoresced **yellow**. Living cell nuclei fluoresced **blue**. Magnification set at 10×.

**Figure 7 ijms-20-04487-f007:**
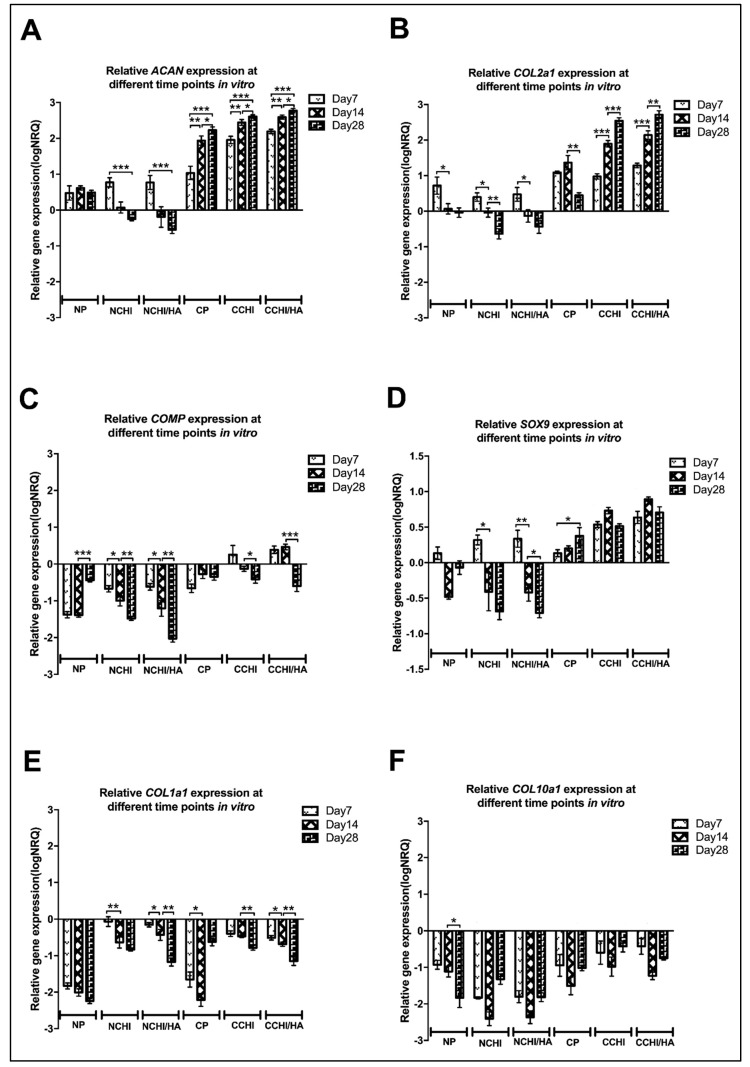
Relative quantitative gene expression of (**A**) *ACAN*, (**B**) *COL2A1*, (**C**) *COMP*, (**D**) *SOX9*, (**E**) *COL1A1*, and (**F**) *COL10A1* in different culture groups (N = normal medium; C = chondrogenic medium, P = 3D Pellet; CHI = chitosan scaffolds, CHI/HA= chitosan with hyaluronic acid scaffold) at different time points (Day 7, 14, and 28) (* *p* < 0.05, ** *p* < 0.01, *** *p* < 0.001). The baseline 0 represents untreated hADSCs in the monolayer, which was the normalization factor.

**Figure 8 ijms-20-04487-f008:**
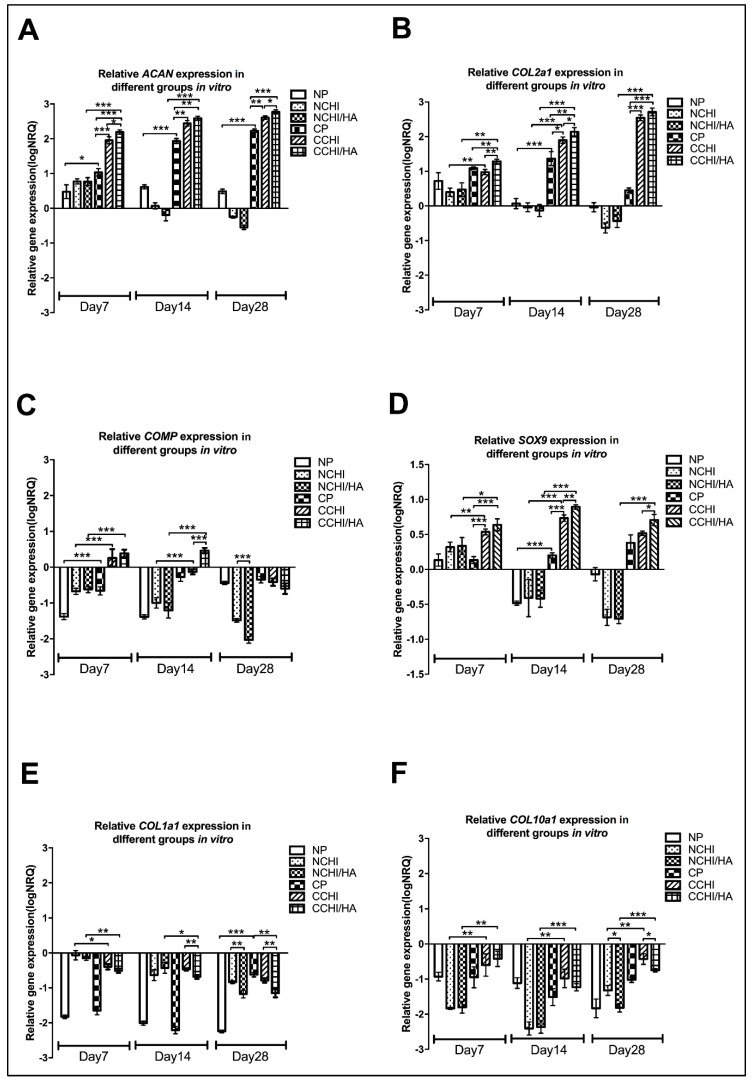
Relative quantitative gene expression of (**A**) *ACAN*, (**B**) *COL2A1*, (**C**) *COMP*, (**D**) *SOX9*, (**E**) *COL1A1*, and (**F**) *COL10A1* on Day 7, 14, and 28 for different culture groups (N = normal medium; C = chondrogenic medium, P = 3D Pellet; CHI= chitosan scaffolds, CHI/HA = chitosan with hyaluronic acid scaffold; * *p* < 0.05, ** *p* < 0.01, *** *p* < 0.001). The baseline 0 represents untreated hADSCs in the monolayer, which was the normalization factor.

**Figure 9 ijms-20-04487-f009:**
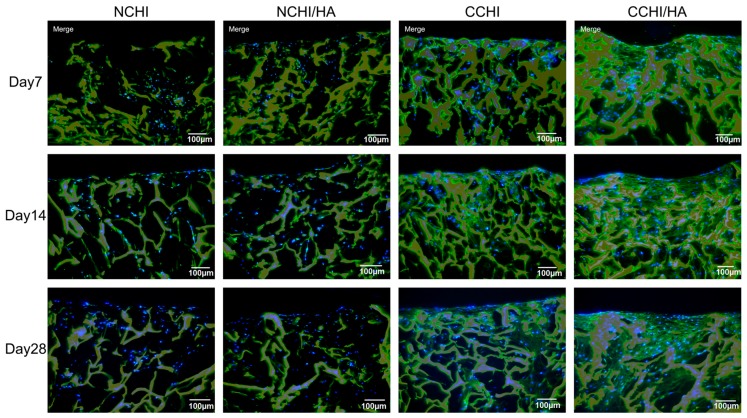
Immunofluorescence staining of collagen type II (**green**) at Day 7, 14, and 28 both in chitosan and chitosan with hyaluronic acid scaffolds with hADSCs cultured in normal (NCHI and NCHI/HA) or chondrogenic medium (CCHI and CCHI/HA). The chitosan and chitosan with hyaluronic acid scaffolds fluoresced **yellow**, whereas living cell nuclei fluoresced **blue.** Magnification set at 10×.

**Table 1 ijms-20-04487-t001:** Gene specific primers used for quantitative real-time PCR.

Gene	Forward Primer (5′–3′)	Reverse Primer (3′–5′)	Accession Nr.	Amplicon Size (bp)
*COL2A1*	GCCCAGTTGGGAGTAAGT	CACCAGGATTGCCTTGAA	NM_001844.4	106
*COL1A1*	GCTGGTCCTCCAGGTGAA	GGGGACCAACAGGACCA	NM_000088.3	159
*COL10A1*	TGGCCTGCCTGACTTTA	AATGTCCAGCTCACTGGA	NM_000493.3	151
*ACAN*	ACCCAAGGACTGGAATCT	CCTGATCCAGGTAGCCTT	NM_001135.3	149
*COMP*	TGCACCGACGTCAACGA	CCGGGTGTTGATGCACA	NM_000095.2	231
*SOX9*	GTGGCTGTAGTAGGAGCT	GCGAACGCACATCAAGA	NM_000346.3	155
*ACTB*	CTGCCCTGAGGCACTC	GTGCCAGGGCAGTGAT	NM_001101.3	197
*RPLP0*	CAACCCAGCTCTGGAGA	CAGCTGGCACCTTATTGG	BC001834.2	116
*TBP*	CACTTCGTGCCCGAAAC	GCCAGTCTGGACTGTTCT	BC110341.1	121
*POLR2e*	CTATCTGGTGACCCAGGA	CTGCAGAAACTGCTCCA	J04965.1	322

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
