# Peer review of "Induction of Articular Chondrogenesis by Chitosan/Hyaluronic-Acid-Based Biomimetic Matrices Using Human Adipose-Derived Stem Cells"

_ijms, 2019, doi:10.3390/ijms20184487_

Round 1
Reviewer 1 Report
Brief Summary: The authors present an in vitro assessment of a biomatrix made by combining chitosan and HA for the in vitro differentiation of adipose-derived tissue MSCs into viable chondrocytes. While the biomatrix itself, made out of chitosan and HA has been reported before for other regenerative applications, its application for articular cartilage regeneration is novel and worth noting. The article is very well written with adequate background information provided in the introduction. I agree with the rationale presented that the adipose stem cells may be a good and more feasible source of regenerative cells. I especially appreciate the detail in which the authors have described their methods and discussed the results, their implications, as well as acknowledge caveats. Below I have a few questions that I would like the authors to address:
What is the size of HA used to prepare the matrix with chitosan?
Low molecular HA in the synovial fluid is recognized as pro-inflammatory and is detected in osteoarthritic knees with cartilage loss. Would the designed biomaterial over time and with degradation cause increased inflammation in the joint? qPCR data for inflammatory genes will in part address this question. Additional studies with macrophages in this matrix will be great.
For the immunofluorescence figures, I would like to see controls with no cells to see the extent to which the antibodies react with or are trapped in the matrices? Images for aggrecan and collagen have very similar staining patterns, is that expected?
The authors do not identify the cell type in the matrices with any cell-marker type analysis (immunofluorescence or qPCR). I would like to see some data identifying the cell type that are generated by differentiation of the ADSCs. Are they only chondrocytes? or more heterogenous with fibroblasts etc.?
Minor comments with regards to typographical errors: On page 2, line 4, the sentence needs correction ".. has also has been..".
Suggestion: On page 4, there is a particularly long and complex sentence, which will be easier to read if broken into separate sentences rather than complexing with "that" multiple times.
With a bit more detail to address the above questions the manuscript may be resubmitted for consideration.
Author Response
Response to Reviewers 1 Comments
We would like to thank the reviewer for his/her response towards our manuscript who clearly identified the critical message that this article conveys. We continuously strive to provide answers and solutions to regenerative and tissue engineering sciences seeking to understand why certain treatments still do not adequately regenerate the relevant tissue types especially bone and cartilage. Articular cartilage regeneration is a key aspect of our research and whilst our manuscript provides some critical answers, more still needs to be done.
All changes to the manuscript have been highlighted in yellow.
Point by point response:
1. What is the size of HA used to prepare the matrix with chitosan?
Response
We thank the reviewer for his/her indication towards the size of the scaffold. The specifics have been added in materials and methods section under page 15 line 19. (15mm in diameter and 8mm in height)
2. Low molecular HA in the synovial fluid is recognized as pro-inflammatory and is detected in osteoarthritic knees with cartilage loss. Would the designed biomaterial over time and with degradation cause increased inflammation in the joint? qPCR data for inflammatory genes will in part address this question. Additional studies with macrophages in this matrix will be great.
Response
We thank the reviewer for his/her comment regarding the pro-inflammatory effect of HA. Whilst the effects are known to us, future studies, as are presently under consideration, will expand on what we have discovered so far. We thank the reviewer in this regard in providing us with further ideas and will incorporate these into our project designs to validate what the intensity of inflammatory reaction could be when a chitosan/HA device is implanted in vivo. However, one must not forget the molecular signals that we have utilized in the study which we believe are the actual driving force behind the articular chondrocyte differentiation from the stem cells. Perhaps it is sufficient to only utilize chitosan with the dual morphogen combination of hBMP-6 with hTGF-b3, thereby eliminating the HA inflammatory aspect entirely?
3. For the immunofluorescence figures, I would like to see controls with no cells to see the extent to which the antibodies react with or are trapped in the matrices? Images for aggrecan and collagen have very similar staining patterns, is that expected?
Response
We thank the reviewer for pointing out this aspect of the immunofluorescence data. Indeed, the chitosan is extremely absorbent which provided a significant hurdle to overcome. We had to considerably play around with the immunofluorescence staining protocol before a desired reaction was achieved that did not cause a total green artifact making matrix and extracellular matrix impossible to distinguish. The trials in the very first sections heavily absorbed the primary and secondary antibodies. Only through considerable adjustments in the staining protocol and playing around with the contrast of the scanned images did we finally achieve images in which the chitosan scaffold had a deep yellowish fluorescent signal with the extracellular matrix fluorescing in green and cells showing a deep blueish tone. This was taken as the standard and was expected to occur in all specimens sections. Future experiments will most likely incorporate alternative fluorescent staining procedures but at the same time shows the challenges scientist face when working with chitosan and should look out for.
4. The authors do not identify the cell type in the matrices with any cell-marker type analysis (immunofluorescence or qPCR). I would like to see some data identifying the cell type that are generated by differentiation of the ADSCs. Are they only chondrocytes? or more heterogenous with fibroblasts etc.?
Response
We thank the reviewer for his/her comment regarding identifying what the stem cells differentiated into. According the presently presented gene expression data we can with a certain degree of certainty say that stem cells under the influence of the chondrogenic medium and in relation to collagen type 2 expression, collagen type 1 and 10 most likely had differentiated into articular chondrogenic-like cells. However, if the same cell type was presented in the scaffolds with normal medium remains an open question that we can at the time being only answer in follow-up experiments and as we expand our gene library of more precise gene markers that would provide clarity on what cells we are actually getting. If all cells were articular or if there was a mixture of fibroblast with chondrocytes our experiment cannot answer at present. Future experiments are considering this and we thank again the reviewer in providing us with ideas on how to expand on our discoveries.
5. Minor comments with regards to typographical errors: On page 2, line 4, the sentence needs correction ".. has also has been..".
6. Suggestion: On page 4, there is a particularly long and complex sentence, which will be easier to read if broken into separate sentences rather than complexing with "that" multiple times.
Response
We thank the reviewer for pointing out critical grammatical errors and sentence structures with extensive length. The manuscript has been extensively overhauled into clear and lucid scientific English including all relevant corrections in this direction.
Reviewer 2 Report
The authors want to make clear whether a chitosan/hyaluronic acid (CHI/HA)matrix supports formation of cartilage with adipose stem cells better than a standard chitosan (CHI)-based construct. They find that without addition of TGF-β and BMP-6, both constructs have very limited capacity to stimulate chondrogenesis and ECM production. This is of clinical relevance (is this new information, or published earlier?) Based on figure 3, they claim that CHI/HA improves stem cell proliferation and viability as compared to CHI. But this claim is incorrect, because the difference between CHI/HA is already present at day 1 and stays the same during 4 weeks culture in both normal and chondrogenic medium. WST-1 and Pico Green assays both reflect cell number. The conclusion should be that in the CHI/HA more cells enter the construct because of its structure, or its increased swelling capacity, or increased binding of stem cells via HA-CD44 interaction, or because the improved hydrophylicity. But all speculations about HA-induced improvement of proliferation and viability should be removed from the Results and Discussion sections.
Furthermore there is a mis-interpretation of the gene-expression results for COL1A1 and COL10A1 (figures 8 and 9). These genes are not downregulated as the authors report in the Results and Discussion sections; they are just lower expressed than the reference genes. In lines 23-24, page 9;"The COL1A1became more downregulated as the culture time increased(Fig. 8E) one could add that COL1A1 expression in the constructs was always much higher than in cell pellets. In lines 39-40, page 9: "expression of COL10A1 and COL1A1 was all down-regulated in both chondrogenic groups". This is not what figures 8 and 9 show:COL1A1 expression at days 14 and 28 is not different in NCHI vs CCHI and in NCHI/HA vs CCHI/HA. Moreover, COL10A1 expression was even highest in the chondrogenic medium cultures CCHI and CCHI/HA! So, there should be no claim that hypertrophy markers are decreased.
By the way, in figures 8 and 9, in the titles there is always "expreesion". Moreover, in the legends the order of the genes studied is almost completely wrong. Only ACAN is at its place.
Page 2, line25: "adipose-derived tissue MSCs" these cells are not mesenchymal.
Page 2, lines 27-30: Based on the the study of Estes et al. you cannot conclude this: they did not compare MSCs and ADSCs, and they compared two passage numbers(4 and 9), but had the differentiation factor TGF-β1 present in the expansion medium. Moreover, increasing chondrogenic differentiation potentials with increasing passage number is contraintuitive and I found no studies confirming this finding.
Immunofluorescence stainings were difficult to interpret. Possibly there are large differences present within one construct? In these stainings there seems to be not that much difference between NCHI, CCHI and CCHI/HA at day 28. I would not expect this, but it was not discussed
Page 9 Statistics: was a correction for multiple comparisons used?.
If the manuscript has been revised, I hope that also the numerous textual errors will be corrected.
Author Response
Response Reviewer 2 Comments
All changes to the manuscript have been highlighted in yellow.
Point by point response:
1. The authors want to make clear whether a chitosan/hyaluronic acid (CHI/HA) matrix supports formation of cartilage with adipose stem cells better than a standard chitosan (CHI)-based construct. They find that without addition of TGF-β and BMP-6, both constructs have very limited capacity to stimulate chondrogenesis and ECM production. This is of clinical relevance (is this new information, or published earlier?) Based on figure 3, they claim that CHI/HA improves stem cell proliferation and viability as compared to CHI. But this claim is incorrect, because the difference between CHI/HA is already present at day 1 and stays the same during 4 weeks culture in both normal and chondrogenic medium. WST-1 and Pico Green assays both reflect cell number. The conclusion should be that in the CHI/HA more cells enter the construct because of its structure, or its increased swelling capacity, or increased binding of stem cells via HA-CD44 interaction, or because the improved hydrophilicity. But all speculations about HA-induced improvement of proliferation and viability should be removed from the Results and Discussion sections.
Response
We would like to thank the reviewer for his/her response towards our manuscript who clearly identified the critical message that this article conveys. We continuously strive to provide answers and solutions to regenerative and tissue engineering sciences seeking to understand why certain treatments still do not adequately regenerate the relevant tissue types especially bone and cartilage. Articular cartilage regeneration is a key aspect of our research and whilst our manuscript provides some critical answers, more still needs to be done.
However, we do not agree fully with the reviewers comments that “Based on figure 3, they claim that CHI/HA improves stem cell proliferation and viability as compared to CHI. But this claim is incorrect, because the difference between CHI/HA is already present at day 1 and stays the same during 4 weeks culture in both normal and chondrogenic medium”. By day 1 there are already apparent differences in cell number and viability, true, but with time there is a clear pattern where the material supports the proliferation and viability of the cells. Indeed, this is an expected result as chitosan and hyaluronic acid both support cell survival and foster proliferation. We do however agree with the reviewer that stating that the chitosan/HA induces proliferation is incorrect. We have as such changed the wording to say “support” as this is better reflects what the material is doing and has been adjusted accordingly in all relevant sections of the manuscript.
2. Furthermore there is a mis-interpretation of the gene-expression results for COL1A1 and COL10A1 (figures 8 and 9). These genes are not downregulated as the authors report in the Results and Discussion sections; they are just lower expressed than the reference genes. In lines 23-24, page 9;"The COL1A1became more downregulated as the culture time increased(Fig. 8E) one could add that COL1A1 expression in the constructs was always much higher than in cell pellets. In lines 39-40, page 9: "expression of COL10A1 and COL1A1 was all down-regulated in both chondrogenic groups". This is not what figures 8 and 9 show:COL1A1 expression at days 14 and 28 is not different in NCHI vs CCHI and in NCHI/HA vs CCHI/HA. Moreover, COL10A1 expression was even highest in the chondrogenic medium cultures CCHI and CCHI/HA! So, there should be no claim that hypertrophy markers are decreased.
Response
We would like to thank the reviewer for his/her comments regarding the gene expression patterns. We have adjusted the manuscript to properly reflect the gene expression patterns and changed up- and down-regulation to more scientifically reflect the gene expression patterns and how they increase or decrease during the experimentation.
However, we do not quite agree with the comment that the hypertrophic marker collagen type X is not decreased and what this implies in the greater context of the results. True “COL1A1 expression at days 14 and 28 is not different in NCHI vs CCHI and in NCHI/HA vs CCHI/HA”, including for COL10A1. The base-line of the gene data represents adipose stem cells which was what normalized to including reference genes (a sentence has been added in the results for clarity as we the materials and methods although containing this are at the end of the manuscript and could have been overlooked). We have used this technique over multiple publications which we direct the reviewer to read for clarity (Klar et al 2014, 2013, and the Ripamonti series after 2014 of publications). We are as such seeing the expressional behavior of “differentiated” cells in relation to a stem cell. From this it becomes clear that the results indicate, depending on culture conditions, that certain genes are highly expressed and others substantially supressed which provides clarity about what the cells have become and are doing. Now in relation to the immunofluorescent data and the gene expression patterns the aspects are fully supported in which the decrease in the gene expression indicates a substantial loss in the production of the molecule. Whilst one can argue on the amount produced, the quantity is too low to be detected and as such can be considered as non-existent. Unfortunately, qPCR data represents this as a negative decrease in the gene expression graqhs which are not the usual fold difference graphs commonly found in the literature. According to the experts in the field that have over numerous publications suggested that representing gene data on a log scale better reflects what the behaviour of the gene. Anything below the null point can be considered as a negative downregulation of the gene expression whereas everything above the line can be considered a positive upregulation. We do acknowledge however that the terminology of up and down-regulation perhaps still warrants better classification and as such have changed this to more scienticially reflect if the the gene is expressed or repressed.
However, despite this the negative decrease in collagen 10 expression patterns can be interpretated as a systematic loss in the expression in Figure 8 and 9 suggesting that the quantitiy of collagen 10, a known hypertrophic marker, is substantially reduced and can be consdered as not being expressed. For a realistic and absolute quantity and the level of loss one would need to do Digital Droplet PCR or Next Gen which in all likelihood would further indicate that the gene is minuscule expressed or better said not expressed. When taken in context with collagen 2 expression patterns the result become clear in which, as we have stated, more research needs to be done to determine what we are getting in the normal medium condition in terms of cells and or matrix.
We have identified certain key points in the discussion which we believe may have contributed to confusion in the message which we have adjusted accordingly.
3. By the way, in figures 8 and 9, in the titles there is always "expreesion". Moreover, in the legends the order of the genes studied is almost completely wrong. Only ACAN is at its place.
Response
We thank the reviewer for pointing this out to us. Figures 8 and 9 have been adjusted accordingly as well as the order of genes in the figure legends to properly reflect with image order in the figures.
4. Page 2, line25: "adipose-derived tissue MSCs" these cells are not mesenchymal.
Response
We thank the reviewer for pointing out this obvious flaw. We have made the relevant changes where appropriate.
5. Page 2, lines 27-30: Based on the the study of Estes et al. you cannot conclude this: they did not compare MSCs and ADSCs, and they compared two passage numbers(4 and 9), but had the differentiation factor TGF-β1 present in the expansion medium. Moreover, increasing chondrogenic differentiation potentials with increasing passage number is contraintuitive and I found no studies confirming this finding.
Response
We thank the reviewer for pointing this out to us. We have re-structured the content to better reflect what was meant to be indicated with the inappropriate reference and content having be removed or restructured where applicable.
6. Immunofluorescence stainings were difficult to interpret. Possibly there are large differences present within one construct? In these stainings there seems to be not that much difference between NCHI, CCHI and CCHI/HA at day 28. I would not expect this, but it was not discussed
Response
We thank the reviewer for their comment but disagree, respectfully, in that there seems to be no difference in immunoflurescent staining at day 28 between collagen type I, collagen type II and ACAN. We are considering doing histomorphometrical assessments in future studies but the best and most accurate software is still under consideration. Present techniques including ImageJ is not sensitive enough to properly reflect these differences in finer detail but apart from that the different images clearly show varying degrees of extracellular matrix and/or cellular material changes with time and shows variations between the different treatment groups. We have adjusted contrast a bit. The density of cells and matrix changes but by how much we cannot say at this time. Page 14 line 12-14 and page 13 line 33-35 deal with this part. Since the staining was challenging we are considering better immunofluorescent stains that better show these differences striving to improve results as more experiments are designed.
7. Page 9 Statistics: was a correction for multiple comparisons used?.
Response
We thank the reviewer for asking about the statistical evaluation. We are considering utilizing Kruskal-Wallis and similar test for multiple comparisons as the scientific community is tending towards standardizing this aspect. We have in this regard not used such a test in the presented study as previous studies by the senior author have shown that these test will only show if a significance is present but where the significane lies between the groups can only be determined by utilizing more focused statistical evaluations, such as the Students t-test. Future experiments are however assessing the feasibility of utilising more relevant tests.
8. If the manuscript has been revised, I hope that also the numerous textual errors will be corrected.
Response
We thank the reviewer for pointing out critical grammatical errors and sentence structures with extensive length. The manuscript has been extensively overhauled into clear and lucid scientific English including all relevant corrections in this direction.
Round 2
Reviewer 1 Report
The subject of this article remains of high interest and is an important first step to demonstrate the application of adipose derived MSCs with Chitosan-HA matrices for articular cartilage regeneration. However, I do not think that the reviewer's comments were sufficiently or satisfactorily addressed by the authors. In my opinion, more information is needed before the manuscript can be accepted.
Author Response
Response to Reviewers Comments
Reviewer 1
Overall response to reviewers comments
I would like to thank the reviewer again for his/her response towards our manuscript who in the first review of the manuscript clearly identified the scientific relevance and the benefit of our work to the scientific knowledge basin.
Below after discussions with the authors we have added additional information into the manuscript that we hope meet with the request and comments made by the reviewer in his first round of reviews:
What is the size of HA used to prepare the matrix with chitosan? Low molecular HA in the synovial fluid is recognized as pro-inflammatory and is detected in osteoarthritic knees with cartilage loss. Would the designed biomaterial over time and with degradation cause increased inflammation in the joint? qPCR data for inflammatory genes will in part address this question. Additional studies with macrophages in this matrix will be great.Response
We have now included additional material in the manuscript in both the introduction and material methods on the two points raised by the reviewer above. This is intended to clarify the nature of the HA used in the experiment which I as the corresponding author originally misinterpreted. I hope the additional material will clarify the issues raised by the reviewer.
For the immunofluorescence figures, I would like to see controls with no cells to see the extent to which the antibodies react with or are trapped in the matrices? Images for aggrecan and collagen have very similar staining patterns, is that expected?
Response
Originally when we started we standardized the immunofluorescent staining using the secondary antibody by omitting the primary to determine background effects. We have attached in the response what the effect looks like from a different IHC staining process where a similar effect is seen with the chitosan and hADSCs. Unfortunately, the controls have since then lost their fluorescent signals and were not recorded by the first author/student who conducted the study. Since the journal has provided us with a limited time to respond to the request and comments by the reviewer, 10 days max as the special issue we are publishing in is set to be published in September 2019, we have insufficient time to redo this to show to the reviewer. However, we would gladly send the reviewer these controls at a later stage such that he/she can convince themselves of our truthfulness in this matter and the challenges we faced.
(PDF attached with pictures)
The authors do not identify the cell type in the matrices with any cell-marker type analysis (immunofluorescence or qPCR). I would like to see some data identifying the cell type that are generated by differentiation of the ADSCs. Are they only chondrocytes? or more heterogenous with fibroblasts etc.?
Response
We have carefully considered this point again. To the best of our knowledge whilst there have been suggestions of possible fibroblast markers these cannot adequately differentiate fibroblasts from chondrocytes. The important aspect is that no bone or adipose tissue matrix was being formed. The cartilage matrix is generally categorized between fibrous and genuine hyaline tissue by its collagen 1 and collagen 10 content, which have tried to show in our gene expression and partially immunofluorescent analysis. As our research progresses we will look for better markers.

Reviewer 2 Report
The authors want to make clear whether a chitosan/hyaluronic acid (CHI/HA) scaffold supports formation of cartilage using adipose stem cells better than a standard chitosan (CHI)-based construct. They find that without addition 0f TGF-β and BMP-6, both constructs have very little capacity to stimulate chondrogenic differentiation of the stem cells and ECM production. This is of clinical relevance, but is this finding new? This dependence on growth factors has been found in stem cells in pellets in earlier studies. If this is really new information, the authors should emphasize this, and otherwise they should refer to the earlier studies that described the finding.
Based on figure 3, the authors claim that CHI/HA improves stem cell proliferation and viability as compared to CHI. But this claim is incorrect, because the difference between CHI/HA and CHI is already present at day 1 and stays the same during 4 weeks culture in both normal and chondrogenic medium; after day 1 the lines of CHI/HA and CHI are running almost parallel in both the WST-1 and picogreen assay.
Moreover, WST-1 and picogreen assays both reflect only the cell number. One can only talk about viability after correction of WST-1 for cell numbers. Moreover, the live/dead survival assay in figure 4 shows that very few dead cells were found in all conditions tested.
The conclusion from figure 3 should be that in the CHI/HA more cells enter the construct because of its structure, or its increased swelling capacity, or increased binding of stem cells via HA-CD44 interaction, or because of improved hydrophilicity. This should be concluded and discussed. All speculations about HA-induced improvement of proliferation and viability/vitality/survivability should be removed from the Results and Discussion sections, because there is no proof in your experiments. This one of the reasons I asked for a major revision, not just the change of the word “induce”s into “supports” at some place, while the old incorrect claim is still mentioned at multiple sites.
For instance in the legends of figure 3 you make such claims. Moreover, in legends of the figures one should only mention what is measured, which groups are compared, the group sizes (n). This means that of the legends of figure 3 only the first 2.5 lines are relevant. In the text of the results part one can describe the effects. In fact this holds for the legends of most of the figures.
The authors are right that I overlooked the sentence in the Materials and Methods section “Data was further normalized to untreated hADSCs in monolayer.” However, to provide clarity the authors have to inform the reader in the Results section and in the Discussion section (e.g. page13, line 33) that mRNA expression of both COL1A1 and COL10A1 is decreased as compared to untreated hADSCs in monolayer culture. Also in the legends of figures 8 and 9 instead of “relative gene expression”, should be replaced by “relative gene expression as compared to untreated hADSCs in monolayer”, because figures should be self-explaining. This will take away this kind of confusion. By the way, the addition the authors made in response to my remarks (Results section, page 8, line 28 “normalization to non-treated non-cultured hADSCs” is different from the text in the Materials and Methods section page 18 “normalized to un-treated hADSCs in monolayer.”What is the real method now?
Another thing about legends: in the legends the authors should not mention results that can not be seen in the corresponding figure! For instance, in the legends of figure 2 they talk about immunofluorescence performed in other experiments.
Another part of my question was ignored. But I ask again: why did the authors not mention, and discuss the fact that COL1A1 expression in the constructs was always much higher than in ADSCs in pellet culture. I also mentioned that the sentence “expression of COL10A1 and COL1A1 was all down-regulated in both modified chondrogenic groups” (now at page 9 , lines 47-48) does not express what figures 8 and 9 show: COL1A1 expression at days 14 and 28 is not different in NCHI vs CCHI and in NCHI/HA vs CCHI/HA. Moreover, COL10A1 expression was even highest in the chondrogenic medium cultures CCHI and CCHI/HA, and even increased during culture! Also, in the Discussion section, page 14, lines 25 -27, the authors make the same mistake. So, there should be no claim that hypertrophy markers are decreased by the chondrogenic medium, like they do on page 9, lines 46-49. Instead, the authors should discuss their unexpected real findings!
In the revised manuscript the authors have replaced “increased” by “increased positively” and also replaced “decreased” by “decreased negatively” or “negatively lower” especially in paragraph 2.6, pages 8 and 9. To my opinion an increase is positive and a decrease is negative, so please undo all these changes.
Page 13, line 33: “ negative gene expression” does not exist. Replace by “decreased gene expression”.
In my remark number 6, I forgot to say that I was looking at the immunofluorescence staining of aggrecan in figure 6. There I see not much difference between staining between NCHI, CCHI and CCHI/HA at day 28. I would not expect this, but it was not discussed. I think the immunofluorescence data are difficult to interpret, also because of large local differences in the structure of the chitosan and the amounts of cells. However, the authors did not mention this
Looking at the immunofluorescence staining of collagen type I in figure 7 it appears to me that here the chitosan background has a different colour, compared to the other figures. As the authors explained to Reviewer 1, they had to play around with the contrast of the scanned images. I think I still see the collagen type I, but that this is difficult because the contrast is too low here. Please comment on this.
Regarding the SEM images in figure 1: without help (arrows?) I don't see the changes the authors describe in the legends and in the results section. Moreover, I expect that there are large local differences in the structure of the chitosan scaffolds. This would mean that it is difficult to choose the relevant pictures, illustrating a process that is taking place.
The answer to my request for correction for multiple comparisons is that you are not going to do that????
Remark 8 regarding the numerous textual errors: I saw that the authors corrected the errors indicated by the two reviewers, but there are many more. Please let someone read the manuscript to correct them all (for instance page 14, line 44: “COMP decreased significantly in the CCHI but decreased only minimally in the CCHI group”, line28 “SOX9 expressing” instead of expression) and to improve the structure of the sentences (for instance page 13, lines 20-24 or lines 48-50; you can’t do this to the reader!). Regarding the response of the authors that the original manuscript has been extensively overhauled into clear and lucid scientific English, I doubt whether the authors want to invest an equal amount of energy in making this manuscript ready for publication as I did. Hopefully the leader of this research group will take his responsibility.
Author Response
Reviewer 2
All changes to the manuscript have been highlighted in yellow.
As the leader, senior and corresponding author of the research group I have always taken responsibility for the research in previous and present groups. I would like to point out to the reviewer that my primary field of experience lies heavily in the bone regeneration. Cartilage regeneration has become an additional field that I am now involving myself in as my research goals are to set the foundations for regenerating lost limbs. With any new venture in new fields of research I am still coming to terms with the basics of cartilage reformation, where I am grateful to this reviewer’s expertise in pointing out aspects that were difficult to interpret. I still believe the material in the article presented, provides important information in which, as all individuals, a unique scientific writing style is present that deviates from other authors. I have tried to consider all of the reviewer’s comments again carefully and have made major amendments to the manuscript to meet the major revision requirements that the reviewer has requested. Also as a native RSA English speaker I together with the second author, also a native English speaker, we have carefully gone through the final changes in which we hope that the English is now clearer.
In regards to what is new and what is not I have included an additional paragraph at the end of article that deals with this and explains some of these aspects. This is however, not what the central aim of the article was but a discovery that we deal with in greater detail in an additional project that aimed to clarify this. The manuscript to this additional Project covers in greater detail the aspect of dual morphogen combinations and their effect on stem cells in scaffolds in vitro which is briefly pointed out in the present manuscript is a novelty that needs in vivo validation.
Point by point response:
Based on figure 3, the authors claim that CHI/HA improves stem cell proliferation and viability as compared to CHI. But this claim is incorrect, because the difference between CHI/HA and CHI is already present at day 1 and stays the same during 4 weeks culture in both normal and chondrogenic medium; after day 1 the lines of CHI/HA and CHI are running almost parallel in both the WST-1 and picogreen assay.Response:
We thank the reviewer for his/her comment. We have made changes in the manuscript accordingly. However, while the Pico Green measure DNA content, thus quantifying cell number directly, WST-1 is influenced by cell number and cell vitality. If the results diverge, cell vitality has changed, i.e. vitality per cell has become different.
Moreover, WST-1 and picogreen assays both reflect only the cell number. One can only talk about viability after correction of WST-1 for cell numbers. Moreover, the live/dead survival assay in figure 4 shows that very few dead cells were found in all conditions tested.
Response
As the reviewer points out this is a vitality test and there are apparent differences between the different culture groups and days. We have made certain amendments in the manuscript to accommodate some of the reviewers request.
The conclusion from figure 3 should be that in the CHI/HA more cells enter the construct because of its structure, or its increased swelling capacity, or increased binding of stem cells via HA-CD44 interaction, or because of improved hydrophilicity. This should be concluded and discussed. All speculations about HA-induced improvement of proliferation and viability/vitality/survivability should be removed from the Results and Discussion sections, because there is no proof in your experiments. This one of the reasons I asked for a major revision, not just the change of the word “induce”s into “supports” at some place, while the old incorrect claim is still mentioned at multiple sites.
Response:
As per the two previous responses we have made certain changes that more subtly reflect what is observed yet maintains part of the message. Whilst the reviewer is correct in stating that in the Live/Dead assay there are few dead cells one can clearly distinguish that there are higher number of cells visually between the various groups and days. It is possible that dead cells may have been removed with medium change. But the images clearly show differences in cell quantities which the other authors also agree upon and where we in part disagree with the reviewer. However, to be diplomatic about the issue I have tried to re-write part of the text to express some of the reviewers points and believe as it is structured properly reflects what needs to be stated.
For instance in the legends of figure 3 you make such claims. Moreover, in legends of the figures one should only mention what is measured, which groups are compared, the group sizes (n). This means that of the legends of figure 3 only the first 2.5 lines are relevant. In the text of the results part one can describe the effects. In fact this holds for the legends of most of the figures. Another thing about legends: in the legends the authors should not mention results that can not be seen in the corresponding figure! For instance, in the legends of figure 2 they talk about immunofluorescence performed in other experiments. Regarding the SEM images in figure 1: without help (arrows?) I don't see the changes the authors describe in the legends and in the results section. Moreover, I expect that there are large local differences in the structure of the chitosan scaffolds. This would mean that it is difficult to choose the relevant pictures, illustrating a process that is taking place.
Response:
I would like to thank the reviewer for his/her comment. Figure legends have been simplified as requested by the reviewer to reflect what they are showing and to not include additional material that is provided in the main body of the text. Only Figure 2-SEM has still some additional information as I believe that it is relevant to point out to the reader what is being seen. In this regard I have also address the reviewers comments about this Figure with respect to guiding arrows. The Figure legend and figure has been modified to show what is supposed to be seen, i.e. cells. As for the geometrical configuration I have briefly described what this is in the figure legend that should now clarifiy this better what to look for.
The authors are right that I overlooked the sentence in the Materials and Methods section “Data was further normalized to untreated hADSCs in monolayer.” However, to provide clarity the authors have to inform the reader in the Results section and in the Discussion section (e.g. page13, line 33) that mRNA expression of both COL1A1 and COL10A1 is decreased as compared to untreated hADSCs in monolayer culture. Also in the legends of figures 8 and 9 instead of “relative gene expression”, should be replaced by “relative gene expression as compared to untreated hADSCs in monolayer”, because figures should be self-explaining. This will take away this kind of confusion. By the way, the addition the authors made in response to my remarks (Results section, page 8, line 28 “normalization to non-treated non-cultured hADSCs” is different from the text in the Materials and Methods section page 18 “normalized to un-treated hADSCs in monolayer.” What is the real method now?
Response:
I thank the reviewer for pointing out this deviation in the normalization. I am presently running various Projects and interpreting data at the same time as I am doing corrections and my endogenous base-line normalization controls are mostly untreated uncultured cells. In the presented manuscript the normalization control was cultured hADSCs in monolayer. The manuscript including the figures now reflects this properly in all areas from results, to figures 8 and 9 including the methods.
Another part of my question was ignored. But I ask again: why did the authors not mention, and discuss the fact that COL1A1 expression in the constructs was always much higher than in ADSCs in pellet culture. I also mentioned that the sentence “expression of COL10A1 and COL1A1 was all down-regulated in both modified chondrogenic groups” (now at page 9 , lines 47-48) does not express what figures 8 and 9 show: COL1A1 expression at days 14 and 28 is not different in NCHI vs CCHI and in NCHI/HA vs CCHI/HA. Moreover, COL10A1 expression was even highest in the chondrogenic medium cultures CCHI and CCHI/HA, and even increased during culture! Also, in the Discussion section, page 14, lines 25 -27, the authors make the same mistake. So, there should be no claim that hypertrophy markers are decreased by the chondrogenic medium, like they do on page 9, lines 46-49. Instead, the authors should discuss their unexpected real findings! In the revised manuscript the authors have replaced “increased” by “increased positively” and also replaced “decreased” by “decreased negatively” or “negatively lower” especially in paragraph 2.6, pages 8 and 9. To my opinion an increase is positive and a decrease is negative, so please undo all these changes. Page 13, line 33: “ negative gene expression” does not exist. Replace by “decreased gene expression”. The answer to my request for correction for multiple comparisons is that you are not going to do that????
Response:
I thank the reviewer for pointing this out again.
Firstly, after having gone through the literature again and also conversing with some of the top qPCR specialists in the field I changed decrease and increases etc back to up-regulation and down-regulation as this is the proper terminology to use when presenting gene fold changes on a logarithmic scale. I please direct the reviewer to read the articles of Goni et al 2009 – The qPCR data statistical analysis including Livak and Schmittgen 2001 – Analysis of relative gene expression data using real-time PCR and the 2(-DeltaDeltaC(T) method AND Remans et al 2014 – Reliable gene expression analysis by qRT-PCR: Reporting and Minimizing the uncertainty in Data Accuracy that deal with this topic. This is a proper reflection of the data and what the gene is doing where there may be increases and decreases in the up and downregulation etc.
In terms of collagen type I and 10 I finally see what the reviewer is pointing out. Yes the reviewer is absolutely correct to state that there is no difference with the chondrogenic medium compared to the normal medium. I have reworded the results and discussion section appropriately to better reflect this. However, it is critical to point out as in my last answer that the results cannot be interpreted based on single genes. The collagen I and 10 have to be taken in consideration to the other cartilage based genes such as collagen 2, Acan and COMP. I have tried to reformulate the discussion appropriately in which the claim that chondrogenic medium eliminates hyperthropy should now be removed. However, chondrogenic medium supports the upregulation of hyaline cartilage based matrix molecules and the discussion still remains structured here etc.
In terms of the correction for multiple comparisons, we did now done this and is represented in the statistical section. Minor changes have occurred but on the whole the statistics remain almost identical to what was done before.
In my remark number 6, I forgot to say that I was looking at the immunofluorescence staining of aggrecan in figure 6. There I see not much difference between staining between NCHI, CCHI and CCHI/HA at day 28. I would not expect this, but it was not discussed. I think the immunofluorescence data are difficult to interpret, also because of large local differences in the structure of the chitosan and the amounts of cells. However, the authors did not mention this Looking at the immunofluorescence staining of collagen type I in figure 7 it appears to me that here the chitosan background has a different colour, compared to the other figures. As the authors explained to Reviewer 1, they had to play around with the contrast of the scanned images. I think I still see the collagen type I, but that this is difficult because the contrast is too low here. Please comment on this.
Response:
I thank the reviewer for specifying what he/she was looking at. With respect to point 13 I cannot see this difference. They all look the same to me and the authors.
In terms of point 12 it is difficult to agree with the reviewer as I can see differences. Indeed the images were difficult to interpret and this is the reason why the qPCR was relevant as it provided some clarity. The aspect of not discussing it was that in the greater context of the discoveries not relevant but we may address these issues in follow up reviews and perspective articles highlighting this more clearly.
Remark 8 regarding the numerous textual errors: I saw that the authors corrected the errors indicated by the two reviewers, but there are many more. Please let someone read the manuscript to correct them all (for instance page 14, line 44: “COMP decreased significantly in the CCHI but decreased only minimally in the CCHI group”, line28 “SOX9 expressing” instead of expression) and to improve the structure of the sentences (for instance page 13, lines 20-24 or lines 48-50; you can’t do this to the reader!). Regarding the response of the authors that the original manuscript has been extensively overhauled into clear and lucid scientific English, I doubt whether the authors want to invest an equal amount of energy in making this manuscript ready for publication as I did. Hopefully the leader of this research group will take his responsibility.
Response:
Dear Reviewer
The last comment of this overall comment is not called for as it is deeply hurtful and in this group leaders eyes highly unprofessional. I strive always to provide the best in terms of scientific English and have taken countless hours/days/weeks to try and correct the content that it fits in terms of grammar and spelling! We all have unique writing styles and I thank the reviewer graciously for pointing out additional errors that I have missed. The second author has also assessed the manuscript and both of us have tried to eliminate further errors. I apologize to the reviewer if my last comments may have caused him/her issues or frustration but we as humans, and prone to errors, can ever only try our best and improve as we grow in the flaws we make.
Round 3
Reviewer 2 Report
The authors have improved the readability of the manuscript and removed most of the wrong conclusions about HA-induced improvement of proliferation and viability.
Here are still some remarks to help the authors improve the manuscript(version with corrections made).
Page 3, line 26: The scaffolds then stabilizes
Page 5, line 1:The difference between (E, G) and (F,H) is not that between cultured and non-cultured scaffolds as you describe here, but only high and low magnification of the same sites of cultured scaffolds!
Page 6, line 6: This indicating that
Page 6, line 24: "On the other hand, only limited fluorescent signals were observed in NCHI and NCHI/HA groups (fig.5, 6). To my opinion this does not hold for the later time-points in the immunofluorescence staining of aggrecan".
Page 8, line 15-17: You exchanged the contents of figure 8 and 9! I would propose the following construction: Relative expression of every gene at different time points but in the same group is shown in Fig. 8, whereas the relative expression of every gene in different groups but at the same time point is shown in Fig. 9.
Page 8, line 18: For clearness, after "up-regulated" I would add ",compared to hADSCs in monolayer,".
Page 8, line 22: ""became down-regulation"?
Page 8, line 26: but not was not"?
Page 9, line 7: For clearness, after "up-regulated" I would add ",compared to hADSCs in monolayer,".
Page 9, line "SOX9 expressed up-regulation"I propose "SOX9 expression was up-regulated".
Page 9, line 11: I propose "became"instead of "become".
Page 9, line 17: "respectably"do you mean "respectively"?
Page 9, lin 18: For clearness, after "down-regulated" I would add ",compared to hADSCs in monolayer,".
Page 9, lines 24-6: here you added (in reaction to my comments) this description of COL1A1 expression in pellet cultures would fit better after "culture condition." in line 19.
After replacing the text according to my previous remark, I would propose that you proceed there with the observation that for both COL1A1 and COL10A1 no benefical effects were found of adding the chondrogenic medium. NCHI vs CCHI and NCHI/HA vs. CCHI/HA were not very different in COL1A1 expression at days 14 and 28, while COL10A1expression was even increased in presence of chondrogenic medium at all time points. I think that it is important to mention this, and to be complete in the description of your results.
Page 9, line 23: "CHI" I would expect that you mean "CCHI" instead.
Page 12, line 30: "after 12 weeks" I propose that you make it äfter 12 weeks in presence of lysozyme", to make clear that this is not spontaneous desintegration of the construct.
Page 12, line 34: I think you better remove "increase in" here, and add in line 36 after "progressed and" the words "this increase".
Page 14, line 6: I would remove "a".
Page 14, line16: For clearness, after "down-regulated" I would add ",compared to hADSCs in monolayer.".
Page 14, line 16: After the addition I proposed in the previous remark, I propose that you, as you were discussing the beneficial effects of the chondrogenic medium on expression of cartilage markers you also discuss here the observation that for both COL1A1 and COL10A1 no benefical effects were found of adding the chondrogenic medium. NCHI vs CCHI and NCHI/HA vs. CCHI/HA were not very different in COL1A1 expression at days 14 and 28, while COL10A1expression was even increased in presence of chondrogenic medium at all time points.
Page 14, lines 35-36: "COMP negatively decreased" I propose that you replace this by ""COMP was down-regulated as compared to unstimulated hADSCs in monolayer".
Page 14, line 38: "As culture time progressed" is not correct, because only day 14 is meant.
Page 14, line 39:"CCHI group" I think you mean "CCHI/HA" group. Moreover, this expression was not "decreased only minimally", but "increased only minimally".
Page 15, line 19:"in a focal defects"?
Author Response
Response to Reviewers Comments
Reviewer 2
All changes to the manuscript have been highlighted in yellow.
Response to the minor corrections please see last paragraph:
"Page 3, line 26: The scaffolds then stabilizes
Page 5, line 1:The difference between (E, G) and (F,H) is not that between cultured and non-cultured scaffolds as you describe here, but only high and low magnification of the same sites of cultured scaffolds!
Page 6, line 6: This indicating that
Page 6, line 24: "On the other hand, only limited fluorescent signals were observed in NCHI and NCHI/HA groups (fig.5, 6). To my opinion this does not hold for the later time-points in the immunofluorescence staining of aggrecan".
Page 8, line 15-17: You exchanged the contents of figure 8 and 9! I would propose the following construction: Relative expression of every gene at different time points but in the same group is shown in Fig. 8, whereas the relative expression of every gene in different groups but at the same time point is shown in Fig. 9.
Page 8, line 18: For clearness, after "up-regulated" I would add ",compared to hADSCs in monolayer,".
Page 8, line 22: ""became down-regulation"?
Page 8, line 26: but not was not"?
Page 9, line 7: For clearness, after "up-regulated" I would add ",compared to hADSCs in monolayer,".
Page 9, line "SOX9 expressed up-regulation"I propose "SOX9 expression was up-regulated".
Page 9, line 11: I propose "became"instead of "become".
Page 9, line 17: "respectably"do you mean "respectively"?
Page 9, lin 18: For clearness, after "down-regulated" I would add ",compared to hADSCs in monolayer,".
Page 9, lines 24-6: here you added (in reaction to my comments) this description of COL1A1 expression in pellet cultures would fit better after "culture condition." in line 19.
After replacing the text according to my previous remark, I would propose that you proceed there with the observation that for both COL1A1 and COL10A1 no benefical effects were found of adding the chondrogenic medium. NCHI vs CCHI and NCHI/HA vs. CCHI/HA were not very different in COL1A1 expression at days 14 and 28, while COL10A1expression was even increased in presence of chondrogenic medium at all time points. I think that it is important to mention this, and to be complete in the description of your results.
Page 9, line 23: "CHI" I would expect that you mean "CCHI" instead.
Page 12, line 30: "after 12 weeks" I propose that you make it äfter 12 weeks in presence of lysozyme", to make clear that this is not spontaneous desintegration of the construct.
Page 12, line 34: I think you better remove "increase in" here, and add in line 36 after "progressed and" the words "this increase".
Page 14, line 6: I would remove "a".
Page 14, line16: For clearness, after "down-regulated" I would add ",compared to hADSCs in monolayer.".
Page 14, line 16: After the addition I proposed in the previous remark, I propose that you, as you were discussing the beneficial effects of the chondrogenic medium on expression of cartilage markers you also discuss here the observation that for both COL1A1 and COL10A1 no benefical effects were found of adding the chondrogenic medium. NCHI vs CCHI and NCHI/HA vs. CCHI/HA were not very different in COL1A1 expression at days 14 and 28, while COL10A1expression was even increased in presence of chondrogenic medium at all time points.
Page 14, lines 35-36: "COMP negatively decreased" I propose that you replace this by ""COMP was down-regulated as compared to unstimulated hADSCs in monolayer".
Page 14, line 38: "As culture time progressed" is not correct, because only day 14 is meant.
Page 14, line 39:"CCHI group" I think you mean "CCHI/HA" group. Moreover, this expression was not "decreased only minimally", but "increased only minimally".
Page 15, line 19:"in a focal defects"?"
I would like to convey my deepest gratefulness to the reviewer, for his/her patience, guidance and support of our manuscript. We have assessed all of the minor comments provided by the reviewer in which we further fine-tuned some aspect of the manuscript in which a point by point response is superficial. We did have difficulties identifying the lines the reviewer was directing us to but we managed to find these all using a few tricks in which I hope that the corrections now meet the reviewers standards.
Thank you again very much and I hope to someday have the privilege of encountering you at a conference to exchange more on the exiting topic.